# TAMING OOD ACTIONS FOR OFFLINE REINFORCEMENT LEARNING: AN ADVANTAGE-BASED APPROACH

## ABSTRACT

Offline reinforcement learning (RL) learns policies from fixed datasets without online interactions, but suffers from distribution shift, causing inaccurate evaluation and overestimation of out-of-distribution (OOD) actions. Existing methods counter this by conservatively discouraging all OOD actions, which limits generalization. We propose Advantage-based Diffusion Actor-Critic (ADAC), which evaluates OOD actions via an advantage-like function and uses it to modulate the Q-function update discriminatively. Our key insight is that the (state) value function is generally learned more reliably than the action-value function; we thus use the next-state value to indirectly assess each action. We develop a PointMaze environment to clearly visualize that advantage modulation effectively selects superior OOD actions while discouraging inferior ones. Moreover, extensive experiments on the D4RL benchmark show that ADAC achieves state-of-the-art performance, with especially strong gains on challenging tasks. Our code is available at `https://anonymous.4open.science/r/adac-14D0`.

## 1 INTRODUCTION

Offline reinforcement learning (RL) (Lange et al., 2012; Levine et al., 2020) focuses on learning decision-making policies solely from previously collected datasets, without online interactions with the environment. This paradigm is particularly appealing for applications where online data collection is prohibitively expensive or poses safety concerns (Kalashnikov et al., 2018; Prudencio et al., 2023). However, offline RL often suffers from distribution shift between the behavior policy and the learned policy. The policy evaluation on out-of-distribution (OOD) actions is prone to extrapolation error (Fujimoto et al., 2019), which can be amplified through bootstrapping, leading to significant overestimation (Kumar et al., 2020).

To mitigate overestimation, a common strategy in offline RL is to incorporate conservatism into algorithm design. Value-based methods (Kumar et al., 2020; Kostrikov et al., 2021; Lyu et al., 2022) achieve this by learning a pessimistic value function that reduces the estimated value of OOD actions to discourage their selection. Alternatively, policy-based methods (Fujimoto & Gu, 2021; Fujimoto et al., 2019; Wang et al., 2022) enforce conservatism by constraining the learned policy to remain close to the behavior policy, thereby avoiding querying OOD actions. Similarly, conditional sequence modeling approaches (Chen et al., 2021; Janner et al., 2022; Ajay et al., 2022) inherently induce conservative behavior by restricting the policy to imitate the behavior contained in the offline dataset. In a distinct manner, model-based approaches (Kidambi et al., 2020; Yu et al., 2020; Sun et al., 2023) ensure conservatism by learning a pessimistic dynamics model where uncertainty-based penalization systematically underestimates the value of OOD actions.

While conservatism is celebrated in offline RL, existing methods achieve it by indiscriminately discouraging all OOD actions, thereby hindering their capacity for generalization. Offline datasets, in practice, are usually characterized by sub-optimal trajectories and narrow state-action coverage. Consequently, an effective offline RL algorithm should possess the ability to stitch sub-optimal trajectories to generate the best possible trajectory supported by the dataset, and even extrapolate beyond the dataset to identify potentially beneficial actions. Such indiscriminate discouragement, however, severely impedes the agent's ability to generalize and achieve high performance. This naturally leads to a fundamental question: *How can we reliably distinguish between undesirable and beneficial OOD actions, and advance the trade-off between conservatism and generalization?*

Figure 1: ADAC Architecture: We use an approximated optimal value function (learned with a batch data) as the measure to evaluate OOD actions. We use an approximate optimal value function (learned from batch data) to evaluate OOD actions. Its relative advantage over in-distribution actions is then used to modulate critic update.

To this end, we propose **A**dvantage-based **D**iffusion **A**ctor-**C**ritic (ADAC), a novel method that more reliably assesses the quality of OOD actions, selectively encourages beneficial ones, and discourages risky ones. Our key insight is that the (state) value function is generally learned more reliably than the action-value function, given limited offline data; we thus use the next-state value to indirectly assess each action. Specifically, we regard an OOD action as advantageous if it can move the current state to a successor state whose value, under the optimal value function, exceeds that of any reachable state under the behavior policy. Since the true optimal value function is inaccessible from offline data, we adopt the dataset-optimal value function (optimal with respect to the offline dataset, see $V_\mu^*$ in Figure 1 and the definition in Eq. (5)) as an approximation. We provide theoretical insights showing that the dataset-optimal value function can be reliably approximated through expectile regression on the dataset. Based on this approximation, we define an advantage function to assess the desirability of actions. It is then used to modulate the temporal difference (TD) target more discriminatively during Q-function (critic) learning. To have a fair comparison, we parameterize the policy (actor) using diffusion models (Ho et al., 2020; Wang et al., 2022) and is updated under the guidance of the learned Q-function. Overall, ADAC evaluates OOD actions more reliably than prior works and achieves state-of-the-art (SOTA) performance on the majority of the D4RL (Fu et al., 2020) benchmark tasks.

## 2 PRELIMINARIES

**Offline Reinforcement Learning.** RL problems are commonly formulated within the framework of a Markov Decision Process (MDP), defined by the tuple $\mathcal{M} = (\mathcal{S}, \mathcal{A}, r, \rho_0, P, \gamma)$. Here, $\mathcal{S}$ denotes the state space, $\mathcal{A}$ represents the action space, and $r(\boldsymbol{s}, \boldsymbol{a}) : \mathcal{S} \times \mathcal{A} \to [-R_{\max}, R_{\max}]$ is a bounded reward function with $R_{\max}$ being the maximum absolute value of the reward. $\rho_0(\boldsymbol{s})$ specifies the initial state distribution, $P(\boldsymbol{s}' \,|\, \boldsymbol{s}, \boldsymbol{a}) : \mathcal{S} \times \mathcal{A} \times \mathcal{S} \to \mathbb{R}_+$ defines the transition dynamics, and $\gamma \in (0, 1)$ is the discount factor (Sutton & Barto, 2018).

A policy $\pi(\cdot|\boldsymbol{s})$ maps a given state $\boldsymbol{s}$ to a probability distribution over the action space. The value function of a state $\boldsymbol{s}$ under a policy $\pi$ is the expected cumulative return when starting in $\boldsymbol{s}$ and following $\pi$ thereafter, i.e., $V^\pi(\boldsymbol{s}) = \mathbb{E}_{\boldsymbol{a_t} \sim \pi(\cdot|\boldsymbol{s_t})} \left[ \sum_{t=0}^\infty \gamma^t r(\boldsymbol{s_t}, \boldsymbol{a_t}) \,\middle|\, \boldsymbol{s_0} = \boldsymbol{s} \right]$, where the expectation takes over the randomness of the policy $\pi$ and the transition dynamics $P$. The optimal state value function $V^*(\cdot)$ satisfies the following Bellman optimality equation:

$$V^*(\boldsymbol{s}) = \max_{\boldsymbol{a} \in \mathcal{A}} \left[ r(\boldsymbol{s}, \boldsymbol{a}) + \gamma \mathbb{E}_{\boldsymbol{s}' \sim P(\cdot \,|\, \boldsymbol{s}, \boldsymbol{a})} V^*(\boldsymbol{s}') \right]. \tag{1}$$

The action-value function (Q-function) is the expected cumulative return when starting from state $\boldsymbol{s}$, taking action $\boldsymbol{a}$, and following $\pi$ thereafter: $Q^\pi(\boldsymbol{s}, \boldsymbol{a}) = \mathbb{E}\left[ \sum_{t=0}^\infty \gamma^t r(\boldsymbol{s_t}, \boldsymbol{a_t}) \,\middle|\, \boldsymbol{s_0} = \boldsymbol{s}, \boldsymbol{a_0} = \boldsymbol{a} \right]$. The goal of RL is to learn a policy $\pi(\cdot \,|\, \boldsymbol{s})$ that maximizes the following expected cumulative long-term reward:

$$J(\pi) = \int_\mathcal{S} \rho_0(\boldsymbol{s}) V^\pi(\boldsymbol{s}) \, ds = \mathbb{E}_{\boldsymbol{s_0} \sim \rho_0, \boldsymbol{a_t} \sim \pi, \boldsymbol{s_{t+1}} \sim P} \left[ \sum_{t=0}^\infty \gamma^t r(\boldsymbol{s_t}, \boldsymbol{a_t}) \right]. \tag{2}$$

As shown in Eq. (2), the classical RL framework requires online interactions with the environment $P$ during training. In contrast, the offline RL learns only from a fixed dataset $\mathcal{D} = \{(\boldsymbol{s}, \boldsymbol{a}, r, \boldsymbol{s}')\}$ collected by the behavior policy $\mu(\cdot \,|\, \boldsymbol{s})$, where $\boldsymbol{s}$, $\boldsymbol{a}$, $r$, and $\boldsymbol{s}'$ denote the state, action, reward, and next state, respectively. That is, it aims to find the best possible policy solely from $\mathcal{D}$ without additional interactions with the environment.

**Diffusion Model.** Diffusion models (Sohl-Dickstein et al., 2015; Ho et al., 2020; Song et al., 2020) consist of a forward process that corrupts data with noise and a reverse process that reconstructs data from noise. Specifically, the forward process is conducted by gradually adding Gaussian noise to samples $\boldsymbol{x_0}$ from an unknown data distribution $p_\theta(\boldsymbol{x_0})$, formulated as:

$$q(\boldsymbol{x_{1:T}} \,|\, \boldsymbol{x_0}) := \prod_{t=1}^{T} q(\boldsymbol{x_t} \,|\, \boldsymbol{x_{t-1}}), \quad q(\boldsymbol{x_t} \,|\, \boldsymbol{x_{t-1}}) := \mathcal{N}(\boldsymbol{x_t}; \sqrt{1 - \beta_t}\boldsymbol{x_{t-1}}, \beta_t\boldsymbol{I}), \tag{3}$$

where $T$ denotes the total number of diffusion steps, and $\beta_t$ controls the variance of the added noise at each step $t$. The reverse process is modeled as $p_\theta(\boldsymbol{x_{0:T}}) := p(\boldsymbol{x_T}) \prod_{t=1}^{T} p_\theta(\boldsymbol{x_{t-1}} \,|\, \boldsymbol{x_t})$, and is trained by maximizing the evidence lower bound (ELBO) (Blei et al., 2017): $\mathbb{E}_q \left[ \log \frac{p_\theta(\boldsymbol{x_{0:T}})}{q(\boldsymbol{x_{1:T}}|\boldsymbol{x_0})} \right]$. After training, samples can be generated by first drawing $\boldsymbol{x_T} \sim p(\boldsymbol{x_T})$ and then sequentially applying the learned reverse transitions to obtain $\boldsymbol{x_0}$. For conditional generation tasks, the reverse process can be extended to model $p_\theta(\boldsymbol{x_{t-1}} \,|\, \boldsymbol{x_t}, c)$, where $c$ denotes the conditioning information.

**Expectile Regression.** Expectile regression has been extensively studied in econometrics (Newey & Powell, 1987) and recently introduced in offline RL (Kostrikov et al., 2021). The $\tau$-expectile (with $\tau \in (0, 1)$) of a real-valued random variable $x$ is defined as the solution to the asymmetric least squares problem:

$$\arg\min_{y\in\mathbb{R}} \mathbb{E}_x \big[ L_2^\tau(x - y) \big], \tag{4}$$

where $L_2^\tau(u) = |\tau - \mathbb{1}(u < 0)|u^2$. Therefore, $\tau = 0.5$ corresponds to the standard mean squared error loss, while $\tau > 0.5$ downweights the contributions of $x$ values smaller than $y$ and assigns greater weight to larger values. Note that as $\tau \to 1$, the solution to Eq. (4) asymptotically approaches the maximum value within the support of $x$.

# 3 THEORETICAL INSIGHT FOR ADVANTAGE-BASED EVALUATION

Our high-level insight is that in offline learning based on limited dataset, the (state) value function is generally learned more reliably than the action-value function, since the data of the latter is a subset of the former. Therefore, we can better evaluate each action indirectly using the next-state value. We start with approximating the optimal value function in the subsection below.

## 3.1 DATASET-OPTIMAL VALUE FUNCTION

Since we only have limited data, we define the following *dataset-optimal value function* (Lyu et al., 2022):

$$V_\mu^*(\boldsymbol{s}) := \max_{\boldsymbol{a}\sim\mu(\cdot|\boldsymbol{s})} \big[ r(\boldsymbol{s}, \boldsymbol{a}) + \gamma\mathbb{E}_{\boldsymbol{s'}\sim P(\cdot|\boldsymbol{s},\boldsymbol{a})} \big[ V_\mu^*(\boldsymbol{s'}) \big] \big]. \tag{5}$$

It differs from the optimal value function Eq. (1) in that it restricts the maximization over actions from behavior policy, that is, the value function of the optimal policy in the dataset. In practice, it can be evaluated by maximizing over sampled data pairs in the offline dataset. Specifically, we adopt expectile regression to *approximate* the maximum operator, while replacing the expectation with empirical samples drawn from the offline dataset $\mathcal{D}$. Then we solve the following regression problem:

$$\mathcal{L}(V) = \mathbb{E}_{(\boldsymbol{s},\boldsymbol{a},r,\boldsymbol{s'})\sim\mathcal{D}} \big[ L_2^\tau(r(\boldsymbol{s}, \boldsymbol{a}) + \gamma V(\boldsymbol{s'}) - V(\boldsymbol{s})) \big]. \tag{6}$$

The minimizer of $\mathcal{L}(V)$ is characterized in the following proposition.

**Proposition 3.1.** *The minimizer $V_\tau(\boldsymbol{s})$ of Eq. (6) is given by*

$$V_\tau(\boldsymbol{s}) = \mathbb{E}_{\boldsymbol{a}\sim\mu(\cdot|\boldsymbol{s}),\ \boldsymbol{s'}\sim P(\cdot|\boldsymbol{s},\boldsymbol{a})}^\tau \big[ r(\boldsymbol{s}, \boldsymbol{a}) + \gamma V_\tau(\boldsymbol{s'}) \big], \tag{7}$$

*where $\mathbb{E}_x^\tau[x]$ denotes the $\tau$-expectile of a real-valued random variable $x$.*

The next proposition characterizes how $V_\tau(\boldsymbol{s})$ approximates the dataset-optimal value function $V_\mu^*(\boldsymbol{s})$.

**Proposition 3.2.** *The solution $V_\tau(\boldsymbol{s})$ to Eq. (6) is uniformly bounded and monotonically non-decreasing with respect to $\tau$. Furthermore, $V_\tau(\boldsymbol{s}) \to \bar{V}(\boldsymbol{s})$ pointwisely as $\tau \to 1$, where $\bar{V}(\boldsymbol{s})$ is given by*

$$\bar{V}(\boldsymbol{s}) = \max_{a\sim\mu(\cdot|\boldsymbol{s})} \left[ r(\boldsymbol{s}, \boldsymbol{a}) + \gamma \max_{\boldsymbol{s'}\sim P(\cdot|\boldsymbol{s},\boldsymbol{a})} \bar{V}(\boldsymbol{s'}) \right]. \tag{8}$$

*In the case of deterministic transition dynamics, the limit coincides with the dataset-optimal value function Eq. (5), i.e., $\bar{V}(\boldsymbol{s}) = V_\mu^*(\boldsymbol{s})$.*

Proposition 3.2 shows that for deterministic transition dynamics, $V_\tau(\boldsymbol{s})$ converges to $V_\mu^*(\boldsymbol{s})$ as $\tau \to 1$. For generic stochastic transition dynamics, since the limit $\bar{V}(\boldsymbol{s})$ involves a maximization over next states (see Eq. (8)), it is possible that $\bar{V}(\boldsymbol{s}) \geq V_\mu^*(\boldsymbol{s})$. In practice, given that $V_\tau(\boldsymbol{s})$ is monotonically non-decreasing in $\tau$, we have achieve a good approximation of $V_\mu^*(\boldsymbol{s})$ by choosing some $\tau < 1$.

Therefore, by solving the regression problem Eq. (6), we can approximate the (dataset-)optimal value function. It is subsequently used to evaluate the quality of OOD actions indirectly.

## 3.2 A New Advantage Function

Our idea is based on the observation that the quality of an action can be better assessed by whether it transitions to a next state with higher value. Specifically, denote the learned value function from solving Eq. (6) by $V(\boldsymbol{s})$, and we define the *advantage* of an action $\boldsymbol{a}$ (possibly OOD) over the behavior policy at state $\boldsymbol{s}$ as

$$A(\boldsymbol{a}|\boldsymbol{s}) := \mathbb{E}_{\boldsymbol{s}' \sim P(\cdot|\boldsymbol{s},\boldsymbol{a})} V(\boldsymbol{s}') - \text{Quantile}_\kappa \left( \left\{ \mathbb{E}_{\boldsymbol{s}_i' \sim P(\cdot|\boldsymbol{s},\boldsymbol{a}_i)} V(\boldsymbol{s}_i') \right\}_{i=1}^N \right), \quad \boldsymbol{a}_i \sim \mu(\cdot|\boldsymbol{s}), \quad (9)$$

where $\{\boldsymbol{a}_i\}_{i=1}^N$ are $N$ actions independently sampled from the behavior policy $\mu(\cdot|\boldsymbol{s})$. In our experiments, we fix $N \equiv 25$ to balance performance and computational efficiency. $\text{Quantile}_\kappa(\cdot)$ denotes the $\kappa$-th quantile of the expected next-state values induced by behavior policy actions.

**Remark.** The newly-defined advantage function is different from the more common one $A(s,a) := Q(s,a) - V(s)$ which employs both the Q-function and V-function. However, relying on Q-function can typically cause over-estimation. In the offline setting, V-function is generally learned more reliably than the action-value function, which can provide better evaluation of an action indirectly.

Under this definition, an action $\boldsymbol{a}$ is considered advantageous if it leads to a next state with a higher expected value than the selective threshold defined by the $\kappa$-th quantile. A positive advantage indicates that the action is favored, while a negative advantage indicates that the action is penalized. The parameter $\kappa$ controls the *level of conservatism*: larger values lead to higher thresholds and encourage conservatism, while smaller values promote optimism by more readily rewarding unseen actions. Notably, all the components in Eq. (9) are learned solely from the offline dataset.

Building on the preceding development, we now augment the standard Bellman operator using the advantage function. Specifically, we introduce the following *advantage-based Bellman operator*:

$$\mathcal{T}_A^{\pi_\theta} Q(\boldsymbol{s}, \boldsymbol{a}) = r(\boldsymbol{s}, \boldsymbol{a}) + \gamma \mathbb{E}_{\boldsymbol{s}' \sim P(\cdot|\boldsymbol{s},\boldsymbol{a}),\, \boldsymbol{a}' \sim \pi_\theta} \left[ Q(\boldsymbol{s}', \boldsymbol{a}') + \lambda A(\boldsymbol{a}'|\boldsymbol{s}') \right], \quad (10)$$

where $\lambda$ is a scaling coefficient that modulates the influence of the advantage function.

Offline RL algorithms based on the standard Bellman backup suffer from action distribution shift during training. This shift arises because the target values in Eq. (10) use actions sampled from the learned policy $\pi_\theta$, while the Q-function is trained only on actions sampled from the behavior policy that produced the dataset ($(\boldsymbol{s}, \boldsymbol{a}) \in \mathcal{D}$). By augmenting the standard Bellman operator with the advantage term $A(\boldsymbol{a}'|\boldsymbol{s}')$, we can effectively mitigate estimation errors in Q-function at OOD actions.

Moreover, we can show that the advantage-based Bellman operator $\mathcal{T}_A^{\pi_\theta}$ is still contractive.

**Proposition 3.3.** *$\mathcal{T}_A^{\pi_\theta}$ is $\gamma$-contractive with respect to the $L_\infty$ norm, which has a unique fixed point.*

We denote the unique fixed point of Eq. (10) by $Q_{\pi_\theta}^A$, and the normal Q-function of $\pi_\theta$ by $Q_{\pi_\theta}$. The following proposition provides the bound of their difference.

**Proposition 3.4.** *The unique fixed point $Q_{\pi_\theta}^A$ of the advantage-based Bellman operator satisfies*

$$\|Q_{\pi_\theta}^A - Q_{\pi_\theta}\|_\infty \leq 2\lambda R_{\max}(1-\gamma)^{-2}.$$

---

**Algorithm 1** Advantage-Based Diffusion Actor-Critic

---

1: **Input** Offline dataset $\mathcal{D}$, policy network $\pi_\theta$, critic networks $Q_\phi$
2: Train a behavior policy $\mu$ by minimizing Eq. (11)
3: Train a value function $V$ by minimizing Eq. (12)
4: Train a transition model $P$ by minimizing Eq. (13)
5: **for** each iteration **do**
6:   Obtain $A(\boldsymbol{a'}|\boldsymbol{s'})$ according to Eq. (9)                    *// Advantage Calculation*
7:   Update $\mathcal{L}_{\mathrm{CRITIC}}(\phi)$ according to Eq. (14)                    *// Critic Update*
8:   Update $\mathcal{L}_{\mathrm{ACTOR}}(\theta)$ according to Eq. (15)                    *// Actor Update*
9:   Soft update parameter $\phi$ and $\theta$                    *// Target networks Update*
10: **end for**

---

We further illustrate the effectiveness of our advantage-based evaluation in Figure 2, where different line styles correspond to different evaluation methods. In the offline RL setting, due to the prohibition against interacting with the environment, directly applying standard Bellman backup (dotted line in Figure 2) results in an erroneous Q-function, which tends to overestimate the value of OOD actions and thus leads to an ineffective policy. Meanwhile, conservative evaluation (dashed line) indiscriminately penalizes all OOD actions, resulting in suppressed Q-values across these actions and limiting the learned policy to a sub-optimal policy near the support of the dataset. By contrast, our advantage-based evaluation (solid line) defines an advantage function that effectively modulates the Q-function obtained from Bellman backup, enabling the policy to discover optimal actions even beyond the support of the dataset. This phenomenon is further validated by empirical results on the PointMaze tasks, as shown in Section 5.2.

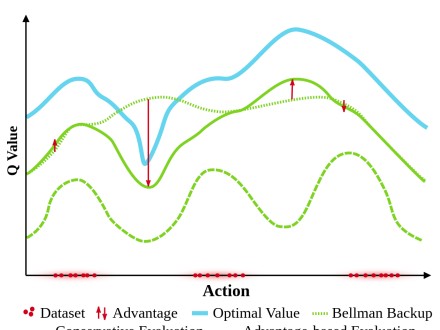

Figure 2: Comparison of prior evaluation methods against our advantage-based evaluation. We visualize the Q-function for a fixed state. The thick blue solid line denotes the optimal value function. The solid line denotes the Q-function learned via advantage-based evaluation. The dashed line denotes the Q-function learned via conservative evaluation. The dotted line denotes the Q-function learned via standard Bellman backup evaluation.

## 4 ADVANTAGE-BASED DIFFUSION ACTOR-CRITIC ALGORITHM

Building on the preceding analysis, we now introduce **A**dvantage-based **D**iffusion **A**ctor-**C**ritic (ADAC).

**Diffusion Policy.** We model our policy as the reverse process of a conditional diffusion model (Wang et al., 2022):

$$\pi_\theta(\boldsymbol{a} \,|\, \boldsymbol{s}) = p_\theta(\boldsymbol{a^{0:T}} \,|\, \boldsymbol{s}) = \mathcal{N}(\boldsymbol{a^T}; \boldsymbol{0}, \boldsymbol{I}) \prod_{i=1}^{T} p_\theta(\boldsymbol{a^{i-1}} \,|\, \boldsymbol{a^i}, \boldsymbol{s}),$$

where the terminal sample $a^0$ is used as the action for RL evaluation. During training, we sample $(\boldsymbol{s}, \boldsymbol{a})$ pairs from the offline dataset $\mathcal{D}$ and construct noisy samples $\boldsymbol{a^i} = \sqrt{\bar{\alpha}_i}\boldsymbol{a} + \sqrt{1 - \bar{\alpha}_i}\boldsymbol{\epsilon}$ (Eq. (3)), where $\alpha_i = 1 - \beta_i$, $\bar{\alpha}_i = \prod_{j=1}^{i} \alpha_j$, and $\boldsymbol{\epsilon} \sim \mathcal{N}(\boldsymbol{0}, \boldsymbol{I})$. Following DDPM (Ho et al., 2020), we train the following noise prediction model $\boldsymbol{\epsilon}_\theta(\boldsymbol{a^i}, \boldsymbol{s}, i)$ to approximate the added noise, which determines the reverse process $p_\theta(\boldsymbol{a^{i-1}} \,|\, \boldsymbol{a^i}, \boldsymbol{s})$:

$$\mathcal{L}_{\mathrm{BC}}(\theta) = \mathbb{E}_{i \sim \mathcal{U}, \boldsymbol{\epsilon} \sim \mathcal{N}(\boldsymbol{0}, \boldsymbol{I}), (\boldsymbol{s}, \boldsymbol{a}) \sim \mathcal{D}} \left[ \left\| \boldsymbol{\epsilon} - \boldsymbol{\epsilon}_\theta(\sqrt{\bar{\alpha}_i}\boldsymbol{a} + \sqrt{1 - \bar{\alpha}_i}\boldsymbol{\epsilon}, \boldsymbol{s}, i) \right\|^2 \right], \tag{11}$$

where $\mathcal{U}$ denotes the uniform distribution over $\{1, \ldots, T\}$. $\mathcal{L}_{\mathrm{BC}}(\theta)$ is a behavior cloning (BC) loss, and minimizing it enables the diffusion model to learn the behavior policy $\mu$. At inference time, an action $\boldsymbol{a^0}$ is generated by sampling $a^T \sim \mathcal{N}(\boldsymbol{0}, \boldsymbol{I})$ and iteratively applying the learned reverse process.

**Advantage.** In our practical implementation, the value function is parameterized by a neural network with parameters $\varphi$ and trained by minimizing the following expectile regression loss:

$$\mathcal{L}_{\text{VALUE}}(\varphi) = \mathbb{E}_{(\boldsymbol{s},\boldsymbol{a},r,\boldsymbol{s'})\sim\mathcal{D}}\big[L_2^\tau(r + \gamma V_\varphi(\boldsymbol{s'}) - V_\varphi(\boldsymbol{s}))\big]. \tag{12}$$

We parameterize the transition dynamics using a neural network and learn a deterministic transition model, which we find sufficiently accurate and computationally efficient in practice. The model is trained by minimizing the following mean squared error (MSE) loss:

$$\mathcal{L}_{\text{MODEL}}(\psi) = \mathbb{E}_{(\boldsymbol{s},\boldsymbol{a},\boldsymbol{s'})\sim\mathcal{D}}\big[\|P_\psi(\boldsymbol{s},\boldsymbol{a}) - \boldsymbol{s'}\|^2\big]. \tag{13}$$

Therefore, the behavior policy $\mu$, the value function $V$, and the transition dynamics $P$ can be learned by minimizing Eq. (11), Eq. (12), and Eq. (13), respectively. The advantage function $A(\boldsymbol{a}\,|\,\boldsymbol{s})$ is then computed as defined in Eq. (9). All components are trained jointly using only offline data and then kept fixed during subsequent actor–critic updates, making this stage computationally inexpensive.

**Actor-Critic.** Following the advantage-based Bellman operator defined in Eq. (10), we define the loss for learning Q-function (critic) as:

$$\mathcal{L}_{\text{CRITIC}}(\phi) = \mathbb{E}_{(\boldsymbol{s},\boldsymbol{a},\boldsymbol{s'})\sim\mathcal{D},\boldsymbol{a'}\sim\pi_\theta(\cdot|\boldsymbol{s'})}\Big[\big(r(\boldsymbol{s},\boldsymbol{a}) + \gamma\big(Q_\phi(\boldsymbol{s'},\boldsymbol{a'}) + \lambda A(\boldsymbol{a'}\,|\,\boldsymbol{s'})\big) - Q_\phi(\boldsymbol{s},\boldsymbol{a})\big)^2\Big]. \tag{14}$$

Here, the advantage function $A(\boldsymbol{a}\,|\,\boldsymbol{s})$ acts as an auxiliary correction term, learned once from offline data and held fixed during critic updates. To improve the policy, we incorporate Q-function guidance into the behavior cloning objective, encouraging the model to sample actions with greater estimated values. The resulting policy (actor) objective combines policy regularization and policy improvement:

$$\mathcal{L}_{\text{ACTOR}}(\theta) = \mathcal{L}_{\text{BC}}(\theta) - \alpha\mathbb{E}_{\boldsymbol{s}\sim\mathcal{D},\boldsymbol{a}\sim\pi_\theta}\big[Q_\phi(\boldsymbol{s},\boldsymbol{a})\big]. \tag{15}$$

We summarize our implementation in Algorithm 1. A central feature of our method is the incorporation of $A(\boldsymbol{a}\,|\,\boldsymbol{s})$, which distinguishes it from prior approaches such as DQL (Wang et al., 2022) that rely solely on the standard Bellman backup.

## 5 Experiments

In this section, we begin by evaluating our method on the widely recognized D4RL benchmark (Fu et al., 2020). We then design a dedicated experiment on the D4RL task PointMaze to better visualize ADAC's ability to identify beneficial OOD actions. Finally, we perform an ablation study to dissect the contribution of key components in our method.

**Dataset.** We evaluate our method on four distinct domains from the D4RL benchmark: Gym, AntMaze, Adroit, and Kitchen. The Gym-MuJoCo locomotion tasks are widely adopted and relatively straightforward due to their simplicity and dense reward signals. In contrast, AntMaze presents more challenging scenarios with sparse rewards, requiring the agent to compose suboptimal trajectories to reach long-horizon goals. The Adroit tasks, collected from human demonstrations, involve narrow state-action regions and demand strong regularization to ensure desired performance. Finally, the Kitchen environment poses a multi-task control problem where the agent must sequentially complete four sub-tasks, emphasizing long-term planning and generalization to unseen states.

**Baseline.** We consider a diverse array of baseline methods that exhibits strong results in each domain of tasks. For policy regularization-based method, we compare with the classic BC, TD3+BC (Fujimoto & Gu, 2021), BEAR (Kumar et al., 2019), BRAC (Wu et al., 2019), BCQ (Fujimoto et al., 2019), AWR (Peng et al., 2019), O-RL (Brandfonbrener et al., 2021), and DQL (Wang et al., 2022). For pessimistic value function-based approach, we include CQL (Kumar et al., 2020), IQL (Kostrikov et al., 2021), and REM (Agarwal et al., 2020). For model-based offline RL, we choose MoRel (Kidambi et al., 2020). For the classic online method, we include SAC (Haarnoja et al., 2018). For conditional sequence modeling approaches, we include DT (Chen et al., 2021), Diffuser (Janner et al., 2022), and DD (Ajay et al., 2022). We report the performance of baseline methods either from the best results published in their respective papers or from (Wang et al., 2022).

### 5.1 Benchmark Results

Our method is evaluated on four task domains, with results summarized in Table 1. We also provide domain-specific analysis to highlight key performance characteristics.

Table 1: Normalized average returns on D4RL tasks, averaged over the final 10 evaluations across 4 seeds.

| Gym Tasks | BC | TD3+BC | CQL | IQL | MoRel | DT | Diffuser | DD | DQL | ADAC (Ours) |
|---|---|---|---|---|---|---|---|---|---|---|
| halfcheetah-medium | 42.6 | 48.3 | 44.0 | 47.4 | 42.1 | 42.6 | 44.2 | 49.1 | 51.1 | **58.0±0.3** |
| hopper-medium | 52.9 | 59.3 | 58.5 | 66.3 | **95.4** | 67.6 | 58.5 | 79.3 | 90.5 | 93.5±4.2 |
| walker2d-medium | 75.3 | 83.7 | 72.5 | 78.3 | 77.8 | 74.0 | 79.7 | 82.5 | 87.0 | **87.6±2.0** |
| halfcheetah-medium-replay | 36.6 | 44.6 | 45.5 | 44.2 | 40.2 | 36.6 | 42.2 | 39.3 | 47.8 | **52.5±0.8** |
| hopper-medium-replay | 18.1 | 60.9 | 95.0 | 94.7 | 93.6 | 82.7 | 96.8 | 100.0 | 101.3 | **102.1±1.1** |
| walker2d-medium-replay | 26.0 | 81.8 | 77.2 | 73.9 | 49.8 | 66.6 | 61.2 | 75.0 | 95.5 | **96.0±1.6** |
| halfcheetah-medium-expert | 55.2 | 90.7 | 91.6 | 86.7 | 53.3 | 86.8 | 79.8 | 90.6 | 96.8 | **106.1±1.0** |
| hopper-medium-expert | 52.5 | 98.0 | 105.4 | 91.5 | 108.7 | 107.6 | 107.2 | 111.8 | 111.1 | **112.5±1.0** |
| walker2d-medium-expert | 107.5 | 110.1 | 108.8 | 109.6 | 95.6 | 108.1 | 108.4 | 108.8 | 110.1 | **112.3±0.9** |
| **Average** | 51.9 | 75.3 | 77.6 | 77.0 | 72.9 | 74.7 | 75.3 | 81.8 | 88.0 | **91.2** |
| **AntMaze Tasks** | BC | TD3+BC | CQL | IQL | BEAR | DT | BCQ | O-RL | DQL | ADAC (Ours) |
| antmaze-umaze | 54.6 | 78.6 | 74.0 | 87.5 | 73.0 | 59.2 | 78.9 | 64.3 | 93.4 | **98.2±4.5** |
| antmaze-umaze-diverse | 45.6 | 71.4 | **84.0** | 62.2 | 61.0 | 53.0 | 55.0 | 60.7 | 66.2 | 76.0±9.9 |
| antmaze-medium-play | 0.0 | 10.6 | 61.2 | 71.2 | 0.0 | 0.0 | 0.0 | 0.3 | 76.6 | **86.5±9.8** |
| antmaze-medium-diverse | 0.0 | 3.0 | 53.7 | 70.0 | 8.0 | 0.0 | 0.0 | 0.0 | 78.6 | **88.7±10.2** |
| antmaze-large-play | 0.0 | 0.2 | 15.8 | 39.6 | 6.7 | 0.0 | 6.7 | 0.0 | 46.4 | **69.8±12.4** |
| antmaze-large-diverse | 0.0 | 0.0 | 14.9 | 47.5 | 2.2 | 0.0 | 2.2 | 0.0 | 56.6 | **64.6±12.7** |
| **Average** | 16.7 | 27.3 | 50.6 | 63.0 | 23.7 | 18.7 | 23.8 | 20.9 | 69.6 | **80.6** |
| **Adroit Tasks** | BC | BRAC-v | CQL | IQL | BEAR | REM | BCQ | SAC | DQL | ADAC (Ours) |
| pen-human | 25.8 | 0.6 | 35.2 | 71.5 | -1.0 | 5.4 | 68.9 | 4.3 | 72.8 | **74.4±18.6** |
| pen-cloned | 38.3 | -2.5 | 27.2 | 37.3 | 26.5 | -1.0 | 44.0 | -0.8 | 57.3 | **80.5±14.3** |
| **Average** | 32.1 | -1.0 | 31.2 | 54.4 | 12.8 | 2.2 | 56.5 | 1.8 | 65.1 | **77.5** |
| **Kitchen Tasks** | BC | BRAC-v | CQL | IQL | BEAR | AWR | BCQ | SAC | DQL | ADAC (Ours) |
| kitchen-complete | 33.8 | 0.0 | 43.8 | 62.5 | 0.0 | 0.0 | 8.1 | 15.0 | 84.0 | **87.9±6.7** |
| kitchen-partial | 33.8 | 0.0 | 49.8 | 46.3 | 13.1 | 15.4 | 18.9 | 0.0 | 60.5 | **65.2±7.0** |
| kitchen-mixed | 47.5 | 0.0 | 51.0 | 51.0 | 47.2 | 10.6 | 8.1 | 2.5 | 62.6 | **68.3±5.8** |
| **Average** | 38.4 | 0.0 | 48.2 | 53.3 | 20.1 | 8.7 | 11.7 | 5.8 | 69.0 | **73.8** |

**Results for AntMaze Tasks.** We follow the D4RL evaluation protocol with *normalized* scores, where 100 corresponds to an expert policy (Fu et al., 2020). AntMaze is particularly challenging due to sparse rewards and the prevalence of suboptimal trajectories, which makes controlled exploration of OOD actions crucial. Under these conditions, ADAC delivers over **15%** improvements across all baselines and task variants. We further observe smoother learning curves and fewer training collapses compared to DQL (Figure 3), suggesting that advantage-guided updates help the agent discover and reliably exploit the small set of reward-yielding behaviors despite limited feedback.

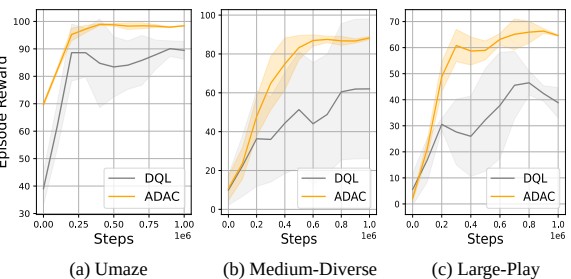

(a) Umaze  (b) Medium-Diverse  (c) Large-Play

Figure 3: Performance comparison of DQL and ADAC on three AntMaze tasks: Umaze, Medium Diverse, and Large Play. Each method was trained with 4 random seeds, and the reward curves were smoothed with a running average ($n = 10$). In this figure, the solid lines correspond to the mean and the shaded regions correspond to the standard deviation.

**Results for Gym Tasks.** In the dense-reward gym mujoco locomotion suite, ADAC provides consistent gains on top of already strong baselines. HalfCheetah is the most challenging family in this suite and exhibits the largest relative improvement at roughly **10%**, while Hopper and Walker2d also benefit. Because scores are normalized (expert = 100, from a converged online policy), averages near or above 90 indicate behavior approaching expert quality; ADAC pushes more tasks into this regime. Qualitatively, we find that the advantage signal helps curb over-optimistic updates on hard-to-model transitions while preserving high-value in-distribution behaviors, yielding both higher final returns and more stable training.

**Results for Adroit and Kitchen Tasks.** Adroit involves dexterous hand manipulation and is particularly prone to extrapolation error because human demonstrations cover a narrow region of the state–action space. While both DQL and ADAC use policy regularization, adding the advantage

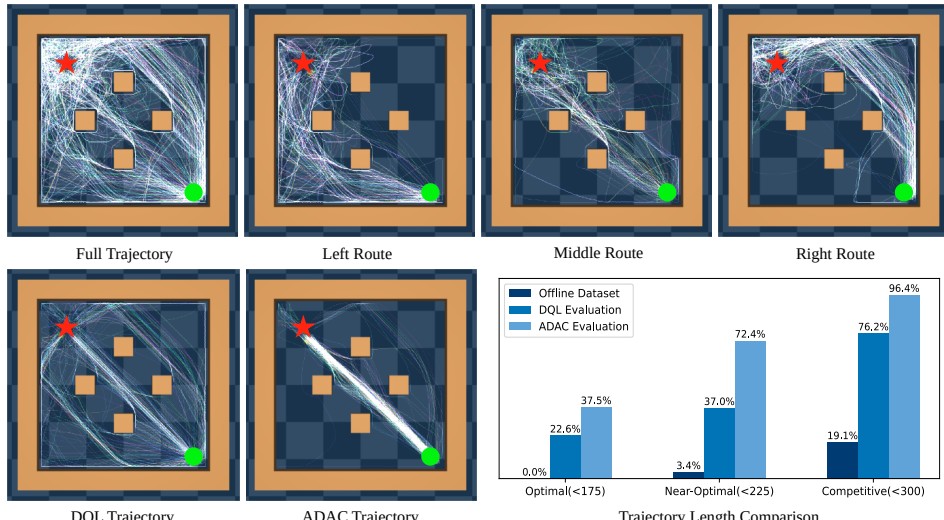

Figure 4: **Sparse reward PointMaze: dataset and method performance. Top:** 853 sub-optimal trajectories are gathered with only terminal reward. Three trajectory patterns—Left (33%), Middle (22%), Right (45%)—span lengths of 200 to 1000 steps (the optimal length is 142). **Bottom:** The first two subfigures illustrate the trajectories generated by DQL and ADAC after training on the dataset, respectively. The last subfigure summarizes the distribution of trajectory lengths for the offline dataset, DQL, and ADAC.

further curbs OOD over-optimism, yielding about **20%** improvement over strong baselines. Kitchen uses a Franka arm to execute long-horizon, compositional goals, and we observe consistent gains there as well.

Together, these results indicate that advantage-guided updates translate beyond locomotion and maze navigation, improving reliability in settings that stress dexterous manipulation and sequential goal completion.

## 5.2 VISUALIZING OOD ACTION SELECTION IN POINTMAZE

In the previous subsection, we demonstrated that our method achieves SOTA performance across a wide range of D4RL benchmark tasks, with particularly large gains on challenging sparse-reward environments. This improvement can be largely attributed to our newly designed advantage function, which enables the selection of beneficial OOD actions—a capability especially critical in sparse-reward tasks.

To better visualize the strength of ADAC in guiding the selection of beneficial OOD actions, we conduct a comparative experiment on PointMaze. Specifically, we construct a toy environment based on the latest `gymnasium-robotics` (de Lazcano et al., 2023) implementation of PointMaze, derived from the Maze2D environment in the D4RL suite. As shown in Figure 4, the green circle indicates the agent's starting position, the red star denotes the goal, and the beige squares represent static obstacles. The task involves navigating a 2-DoF point agent through a maze with obstacles to a fixed goal using Cartesian $(x, y)$ actuation. This is a sparse-reward task: the agent receives a reward of 1 only upon reaching the goal and zero at all other steps. We manually collect 853 trajectories of varying quality, as illustrated in the bar plot of Figure 4, which together yield 391,391 tuples of the form $(s, a, s', r, \text{done})$. Details of the dataset collection strategy and the trajectory quality analysis are provided in Appendix C.7.

We train both DQL (Wang et al., 2022) and ADAC for 50 000 steps using the constructed dataset to obtain their respective policies. Each policy is then evaluated in our environment, generating 300 trajectories per method, as illustrated in the bottom row of Figure 4.

Note that the original collected offline trajectories does not contain any optimal trajectories (of steps less than 175, see the bar plot of Figure 4). Nevertheless, ADAC effectively learns the optimal trajectories from these suboptimal datasets, which generated a substantial number of straight-line

(optimal) trajectories from the start to the goal—routes that are entirely absent in the dataset. This visual evidence strongly supports that ADAC is capable of identifying and selecting superior OOD actions. In contrast, the trajectories generated by DQL, while showing moderate improvement over the offline data, still largely follow left-, middle-, and right-pattern behaviors, indicating a strong tendency toward behavior cloning. Since the key distinction between DQL and ADAC lies in the introduction of advantage modulation, these visualizations clearly validate the effectiveness of the advantage function in enabling the selection of beneficial OOD actions. The last subfigure of Figure 4 provides quantitative evidence that ADAC substantially outperforms DQL in trajectory quality.

## 5.3 ABLATION ON THE ADVANTAGE COMPONENT

We assess the importance of the advantage by comparing models *with* and *without* this component on four representative domains: Gym Locomotion, AntMaze, Adroit, and Kitchen. The ablation results in Figure 5 show that introducing the advantage consistently strengthens performance across all settings, with relative improvements of **11.1%** on Gym Locomotion, **12.2%** on AntMaze, **10.9%** on Adroit, and **12.4%** on Kitchen. Beyond the overall gains, the pattern is consistent across domains with diverse dynamics and reward structures, indicating that the advantage contributes broadly rather than acting as a domain-specific tweak.

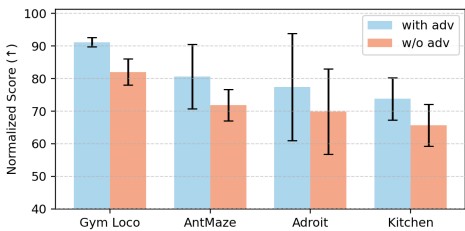

Figure 5: Ablation of the advantage component across four domains. Bars show domain-level mean normalized scores.

## 5.4 COMPUTATIONAL EFFICIENCY OF TRAINING AND INFERENCE

All experiments were conducted on a single NVIDIA RTX 4090, without any distributed training or model parallelism. This setup ensures that reported throughput reflects algorithmic and implementation efficiency rather than scale-out effects.

Building on the lightweight utilities of `jaxrl_m` (Flax/JAX), we reimplemented Diffusion QL in JAX and subsequently introduced advantage-centric components on top of it. The resulting codebase follows a flat, modular design that facilitates reproduction and portability across tasks. Under identical hardware

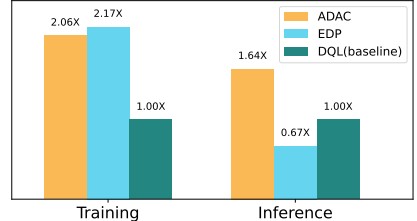

Figure 6: Training and inference speedups of ADAC compared to EDP and the DQL baseline.

and evaluation protocols, the implementation delivers consistent throughput, yielding a $2\times$ improvement in training and a $1.64\times$ improvement in inference over the original DQL implementation. In head-to-head comparisons with Efficient Diffusion Policy (EDP) (Kang et al., 2023), training throughput is comparable, while inference is faster (Fig. 6).

## 6 CONCLUSION

In this work, we propose ADAC, a novel offline RL algorithm that systematically evaluates the quality of OOD actions to balance conservatism and generalization. ADAC represents a pioneering attempt to explicitly assess OOD actions and to selectively encourage beneficial ones, while discouraging risky ones to maintain conservatism. We validate the effectiveness of advantage modulation through a series of custom PointMaze experiments and demonstrate state-of-the-art performance across almost all tasks in the D4RL benchmark. The empirical results further indicate that ADAC is particularly effective in more challenging tasks.

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

# Supplementary Material

## Table of Contents

## A    RELATED WORK

Offline RL focuses on learning effective policies solely from a pre-collected behavior dataset and has demonstrated significant success in practical applications (Rafailov et al., 2021; Singh et al., 2020; Li et al., 2010). The existing literature on offline RL can be classified into four main categories:

**Pessimistic value-based** methods achieve conservatism by incorporating penalty terms into the value optimization objective, discouraging the value function from being overly optimistic on out-of-distribution (OOD) actions. Specifically, CQL (Kumar et al., 2020) applies equal penalization to Q-values for all OOD samples, whereas EDAC (An et al., 2021) and PBRL (Bai et al., 2022) adjust the penalization based on the uncertainty level of the Q-value, measured using a neural network ensemble.

**Regularized policy-based** methods constrain the learned policy to stay close to the behavior policy, thereby avoiding OOD actions. For instance, BEAR (Kumar et al., 2019) constrains the optimized policy by minimizing the MMD distance to the behavior policy. BCQ (Fujimoto et al., 2019) restricts the action space to those present in the dataset by utilizing a learned Conditional-VAE (CVAE) behavior-cloning model. Alternatively, TD3+BC (Fujimoto & Gu, 2021) simply adds a behavioral cloning regularization term to the policy optimization objective and achieves excellent performance across various tasks. IQL (Kostrikov et al., 2021) adopts an advantage-weighted behavior cloning approach, learning Q-value functions directly from the dataset. Meanwhile, DQL (Wang et al., 2022) leverages diffusion policies as an expressive policy class to enhance behavior-cloning. Our work falls into this category as we also incorporate a behavior-cloning term.

**Conditional sequence modeling** methods induce conservatism by limiting the policy to replicate behaviors from the offline dataset (Chen et al., 2021; Wu et al., 2023). This leads to a supervised learning paradigm. Additionally, trajectories can be formulated as conditioned generative models and generated by diffusion models that satisfy conditioned constraints (Janner et al., 2022; Ajay et al., 2022).

**Model-based** methods incorporate conservatism to prevent the policy from overgeneralizing to regions where the dynamics model predictions are unreliable. For example, COMBO (Yu et al., 2021) extends CQL to a model-based setting by enforcing small Q-values for OOD samples generated by the dynamics model. RAMBO (Rigter et al., 2022) incorporates conservatism by adversarially training the dynamics model to minimize the value function while maintaining accurate transition predictions. Most model-based methods achieve conservatism through uncertainty quantification, penalizing rewards in regions with high uncertainty. Specifically, MOPO (Yu et al., 2020) uses the max-aleatoric uncertainty quantifier, MOReL (Kidambi et al., 2020) employs the max-pairwise-diff uncertainty quantifier, and MOBILE (Sun et al., 2023) leverages the Model-Bellman inconsistency uncertainty quantifier. Recently, (Chen et al., 2025) achieves conservatism by incorporating the value function inconsistency loss, enabling the training of a more reliable model.

## B    PROOF OF PROPOSITIONS

In this subsection, we provide comprehensive and complete proofs for our propositions listed in Section 3.

**Proof of Proposition 3.1.**

*Proof.* Denote $\delta(\boldsymbol{s}, \boldsymbol{a}, \boldsymbol{s}') = r(\boldsymbol{s}, \boldsymbol{a}) + \gamma V(\boldsymbol{s}') - V(\boldsymbol{s})$, the regression problem in Eq. (6) can be rewritten as

$$\mathcal{L}(V) = \mathbb{E}_{(\boldsymbol{s}, \boldsymbol{a}, r, \boldsymbol{s}') \sim \mathcal{D}}\big[L_2^\tau(\delta(\boldsymbol{s}, \boldsymbol{a}, \boldsymbol{s}'))\big].$$

To find the optimal $V(\boldsymbol{s})$, we take the derivative of $\mathcal{L}(V)$ with respect to $V(\boldsymbol{s})$ conditioning on $\boldsymbol{s}$:

$$\begin{aligned}
\frac{\partial \mathcal{L}(V)}{\partial V(\boldsymbol{s})} &= \mathbb{E}_{\boldsymbol{a} \sim \mu(\cdot|\boldsymbol{s}), \boldsymbol{s}' \sim P(\cdot|\boldsymbol{s}, \boldsymbol{a})}\left[\frac{\partial L_2^\tau(\delta)}{\partial V(\boldsymbol{s})}\right] \\
&= \mathbb{E}_{\boldsymbol{a} \sim \mu(\cdot|\boldsymbol{s}), \boldsymbol{s}' \sim P(\cdot|\boldsymbol{s}, \boldsymbol{a})}\left[\frac{\partial L_2^\tau(\delta)}{\partial \delta} \cdot \frac{\partial \delta}{\partial V(\boldsymbol{s})}\right] \\
&= \mathbb{E}_{\boldsymbol{a} \sim \mu(\cdot|\boldsymbol{s}), \boldsymbol{s}' \sim P(\cdot|\boldsymbol{s}, \boldsymbol{a})}\big[2|1 - \mathbb{1}(\delta < 0)|\delta(\boldsymbol{s}, \boldsymbol{a}, \boldsymbol{s}') \cdot (-1)\big],
\end{aligned}$$

where the exchange of partial derivative and expectation is due to dominated convergence theorem since both $r$ and $V$ are bounded.

From the fact that the solution $V_\tau(\boldsymbol{s})$ satisfies $\frac{\partial \mathcal{L}(V)}{\partial V(\boldsymbol{s})}\big|_{V_\tau(\boldsymbol{s})} = 0$, we get

$$\mathbb{E}_{\boldsymbol{a} \sim \mu(\cdot|\boldsymbol{s}), \boldsymbol{s}' \sim P(\cdot|\boldsymbol{s}, \boldsymbol{a})}\big[|\tau - \mathbb{1}(r(\boldsymbol{s}, \boldsymbol{a}) + \gamma V_\tau(\boldsymbol{s}') - V_\tau(\boldsymbol{s}) < 0)|(r(\boldsymbol{s}, \boldsymbol{a}) + \gamma V_\tau(\boldsymbol{s}') - V_\tau(\boldsymbol{s}))\big] = 0.$$

In expectile regression, the $\tau$-expectile $\mu_\tau$ of a random variable $X$ satisfies

$$\mathbb{E}\big[|\tau - \mathbb{1}(X - \mu_\tau < 0)|(Y - \mu_\tau)\big] = 0.$$

As a result, this implies that the solution $V_\tau(\boldsymbol{s})$ is the $\tau$-expectile of the target $r(\boldsymbol{s}, \boldsymbol{a}) + \gamma V_\tau(\boldsymbol{s}')$. Therefore, we conclude that

$$V_\tau(\boldsymbol{s}) = \mathbb{E}_{\boldsymbol{a} \sim \mu(\cdot|\boldsymbol{s}), \boldsymbol{s}' \sim P(\cdot|\boldsymbol{s}, \boldsymbol{a})}^\tau\big[r(\boldsymbol{s}, \boldsymbol{a}) + \gamma V_\tau(\boldsymbol{s}')\big],$$

which finish our proof. $\square$

**Proof of Proposition 3.2.**

*Proof.* Define the $\tau$-expectile Bellman operator as

$$\mathcal{T}_\tau V(\boldsymbol{s}) := \mathbb{E}_{\boldsymbol{a} \sim \mu(\cdot\,|\,\boldsymbol{s}), \boldsymbol{s}' \sim P(\cdot\,|\,\boldsymbol{s}, \boldsymbol{a})}^\tau\big[r(\boldsymbol{s}, \boldsymbol{a}) + \gamma V(\boldsymbol{s}')\big].$$

From Eq. (7), we know that $\mathcal{T}_\tau V_\tau(\boldsymbol{s}) = V_\tau(\boldsymbol{s})$, which means $V_\tau(\boldsymbol{s})$ is a fixed point for $\tau$-expectile Bellman operator $\mathcal{T}_\tau$.

Suppose there is another fixed point $W_\tau(\boldsymbol{s})$ for $\mathcal{T}_\tau$. It holds that

$$\|V_\tau - W_\tau\|_\infty = \|\mathcal{T}_\tau V_\tau - \mathcal{T}_\tau W_\tau\|_\infty \leq \gamma\|V_\tau - W_\tau\|_\infty,$$

which means that $V_\tau = W_\tau$. Therefore, we have shown that $V_\tau$ is the unique fixed point for $\mathcal{T}_\tau$.

For $\tau_1 \le \tau_2$ and a bounded random variable $X$, we have

$$\mathbb{E}^{\tau_1}[X] \le \mathbb{E}^{\tau_2}[X].$$

As a result, we get $\mathcal{T}_{\tau_1}V \le \mathcal{T}_{\tau_2}V$. Therefore, for its fixed point, we have $V_{\tau_1} \le V_{\tau_2}$.

It can be shown that

$$V_\tau(\boldsymbol{s}) = \mathcal{T}_\tau V_\tau(\boldsymbol{s}) \le \max_{\boldsymbol{a}\in\mu(\cdot|\text{s})} \big[ r(\boldsymbol{s},\boldsymbol{a}) + \gamma \|V_\tau\|_\infty \big] \le \frac{2R_{\max}}{1-\gamma}.$$

Therefore, we have demonstrated that $V_\tau(\boldsymbol{s})$ is bounded and monotonically non-decreaing in $\tau$. Consequently, there exists a limit $\bar{V}(\boldsymbol{s})$ such that

$$\lim_{\tau\to 1} V_\tau(\boldsymbol{s}) = \bar{V}(\boldsymbol{s}).$$

Define a random variable

$$X^\tau = r(\boldsymbol{s},\boldsymbol{A}) + \gamma V_\tau(\boldsymbol{S}'), \quad \boldsymbol{A} \sim \mu(\cdot\,|\,\boldsymbol{s}), \boldsymbol{S}' \sim P(\cdot\,|\,\boldsymbol{s},\boldsymbol{A}),$$

and its limit $\bar{X} = r(\boldsymbol{s},\boldsymbol{A}) + \gamma \bar{V}(\boldsymbol{S}')$. It follows that

$$\lim_{\tau\to 1} V_\tau(\boldsymbol{s}) = \lim_{\tau\to 1} \mathbb{E}^\tau[X^\tau] \overset{(1)}{=} \lim_{\tau\to 1} \mathbb{E}^\tau[\bar{X}],$$

where (1) comes from

$$|\mathbb{E}^\tau[X^\tau] - \mathbb{E}[\bar{X}]| \le \mathbb{E}|X^\tau - \bar{X}|.$$

From Lemma 1 in (Kostrikov et al., 2021), which states that

$$\lim_{\tau\to 1} \mathbb{E}^\tau[\bar{X}] = \max(\bar{X}).$$

Therefore, we get

$$\bar{V}(\boldsymbol{s}) = \max_{\boldsymbol{a}\in\mu(\cdot|\boldsymbol{s})} \big[ r(\boldsymbol{s},\boldsymbol{a}) + \gamma \max_{\boldsymbol{s}'\sim P} \bar{V}(\boldsymbol{s}) \big].$$

For deterministic transition probability $P$, we have

$$\bar{V}(\boldsymbol{s}) = \max_{\boldsymbol{a}\in\mu(\cdot|\boldsymbol{s})} \big[ r(\boldsymbol{s},\boldsymbol{a}) + \gamma \mathbb{E}_{\boldsymbol{s}'\sim P(\cdot|\boldsymbol{s},\boldsymbol{a})} \bar{V}(\boldsymbol{s}) \big].$$

Define the batch-optimal Bellman operator as

$$\mathcal{T}_\mu^* V(\boldsymbol{s}) = \max_{\boldsymbol{a}\in\mu(\cdot|\boldsymbol{s})} \big[ r(\boldsymbol{s},\boldsymbol{a}) + \gamma \mathbb{E}_{\boldsymbol{s}'\sim P(\cdot|\boldsymbol{s},\boldsymbol{a})} V(\boldsymbol{s}) \big].$$

It follows that $\bar{V}(\boldsymbol{s})$ and $V_\mu^*(\boldsymbol{s})$ are both fixed point for $\mathcal{T}_\mu^*$. By a similar argument for $\mathcal{T}_\tau$, we know that $\mathcal{T}_\mu^*$ is $\gamma$-contractive and has a unique fixed point. As a result, it holds that

$$\lim_{\tau\to 1} V_\tau(\boldsymbol{s}) = V_\mu^*(\boldsymbol{s})$$

for a deterministic transition probability. Overall, we finish our proof. $\qquad\square$

**Proof of Proposition 3.3.**

*Proof.* Let $Q_1$ and $Q_2$ be two arbitrary $Q$-functions. We have

$$\begin{aligned}
\|\mathcal{T}_A^{\pi_\theta} Q_1 - \mathcal{T}_A^{\pi_\theta} Q_2\|_\infty &= \max_{\boldsymbol{s},\boldsymbol{a}} \big| r(\boldsymbol{s},\boldsymbol{a}) + \gamma \mathbb{E}_{\boldsymbol{s}'\sim P, \boldsymbol{a}'\sim\pi_\theta} \big[ Q_1(\boldsymbol{s}',\boldsymbol{a}') + \lambda A(\boldsymbol{a}'|\boldsymbol{s}') \big] \\
&\qquad - r(\boldsymbol{s},\boldsymbol{a}) - \gamma \mathbb{E}_{\boldsymbol{s}'\sim P, \boldsymbol{a}'\sim\pi_\theta} \big[ Q_2(\boldsymbol{s}',\boldsymbol{a}') + \lambda A(\boldsymbol{a}'|\boldsymbol{s}') \big] \big| \\
&= \max_{\boldsymbol{s},\boldsymbol{a}} \big| \gamma \mathbb{E}_{\boldsymbol{s}'\sim P, \boldsymbol{a}'\sim\pi_\theta} \big[ Q_1(\boldsymbol{s}',\boldsymbol{a}') - Q_2(\boldsymbol{s}',\boldsymbol{a}') \big] \big| \\
&\le \gamma \max_{\boldsymbol{s},\boldsymbol{a}} \|Q_1 - Q_2\|_\infty \\
&= \gamma \|Q_1 - Q_2\|_\infty.
\end{aligned}$$

Therefore, $\mathcal{T}_A^{\pi_\theta}$ is a $\gamma$-contraction operator which naturally implies any initial Q-function can converge to a unique fixed point by repeatedly applying this operator. $\qquad\square$

**Proof of Proposition 3.4.**

*Proof.* We first show that $\|A(\boldsymbol{a}'|\boldsymbol{s}')\|_\infty \leq 2R_{\max}/(1-\gamma)$. From the proof of Theorem 3.2, we know that

$$\mathcal{T}_\tau V(\boldsymbol{s}) := \mathbb{E}^\tau_{\boldsymbol{a} \sim \mu(\cdot\,|\,\boldsymbol{s}), \boldsymbol{s}' \sim P(\cdot\,|\,\boldsymbol{s}, \boldsymbol{a})} \big[ r(\boldsymbol{s}, \boldsymbol{a}) + \gamma V(\boldsymbol{s}') \big].$$

Since the $\tau$-expectile of a random variable cannot exceed its maximum, for any $V$, we have

$$\|\mathcal{T}_\tau V\|_\infty \leq R_{\max} + \gamma \|V\|_\infty.$$

From Eq. (7), we know that

$$\|V_\tau\|_\infty = \|\mathcal{T}_\tau V_\tau\|_\infty \leq R_{\max} + \gamma \|V_\tau\|_\infty.$$

It follows that

$$\|V_\tau\|_\infty \leq \frac{R_{\max}}{1-\gamma}, \forall \tau.$$

Therefore, we get $\|V\|_\infty \leq R_{\max}/(1-\gamma)$ and $\|A(\boldsymbol{a}'|\boldsymbol{s}')\|_\infty \leq 2R_{\max}/(1-\gamma)$.

From the advantage-based operator $\mathcal{T}_A^{\pi_\theta}$, we have

$$\mathcal{T}_A^{\pi_\theta} Q(\boldsymbol{s}, \boldsymbol{a}) = r(\boldsymbol{s}, \boldsymbol{a}) + \gamma \mathbb{E}_{\boldsymbol{s}' \sim P, \boldsymbol{a}' \sim \pi_\theta} \bigg[ Q(\boldsymbol{s}', \boldsymbol{a}') + \lambda A(\boldsymbol{a}'|\boldsymbol{s}') \bigg]$$

$$= r(\boldsymbol{s}, \boldsymbol{a}) + \gamma \mathbb{E}_{\boldsymbol{s}' \sim P, \boldsymbol{a}' \sim \pi_\theta} \bigg[ Q(\boldsymbol{s}', \boldsymbol{a}') \bigg] + \gamma \mathbb{E}_{\boldsymbol{s}' \sim P, \boldsymbol{a}' \sim \pi_\theta} \big[ \lambda A(\boldsymbol{a}'|\boldsymbol{s}') \big]$$

$$= \mathcal{T}^{\pi_\theta} Q(\boldsymbol{s}, \boldsymbol{a}) + \gamma \mathbb{E}_{\boldsymbol{s}' \sim P, \boldsymbol{a}' \sim \pi_\theta} \big[ \lambda A(\boldsymbol{a}'|\boldsymbol{s}') \big],$$

where $\mathcal{T}^{\pi_\theta}$ is the standard Bellman operator. From the boundedness of $A(\boldsymbol{a}|\boldsymbol{s})$, we have

$$\mathcal{T}^{\pi_\theta} Q(\boldsymbol{s}, \boldsymbol{a}) - \gamma \frac{2\lambda R_{\max}}{1-\gamma} \leq \mathcal{T}_A^{\pi_\theta} Q(\boldsymbol{s}, \boldsymbol{a}) \leq \mathcal{T}^{\pi_\theta} Q(\boldsymbol{s}, \boldsymbol{a}) + \gamma \frac{2\lambda R_{\max}}{1-\gamma}.$$

Iteratively applying this operator to obtain the fixed point, we get

$$Q_{\pi_\theta} - \frac{2\lambda R_{\max}}{(1-\gamma)^2} \leq Q_{\pi_\theta}^A \leq Q_{\pi_\theta} + \frac{2\lambda R_{\max}}{(1-\gamma)^2}, \forall \boldsymbol{s}, \boldsymbol{a},$$

which implies our conclusion. $\square$

## C  Experimental Details

### C.1  Advantage Function Characterization and Regularization

To provide an overview of the learned advantage function, we report summary statistics computed across 20 D4RL tasks using the best-performing hyperparameter setting (Appendix C.4). For each task, we aggregate advantage values from four independent training runs and report: (1) the mean and standard deviation of both positive and negative advantages, and (2) the proportion of samples exhibiting positive advantages.

The results in Table 2 reveal notable variability in the distribution of advantage values across tasks. In particular, the proportion of positive advantages differs substantially between environments, reflecting how often the learned value function favors alternative actions over those observed in the dataset. We observe that some tasks display a low proportion of positive advantages—reflecting fewer opportunities for improvement—whereas others show substantially higher positive ratios, indicating greater diversity in action quality and more room for enhancement. These statistics are determined by factors such as the quality of the behavior policy, hyperparameters like $\kappa$, and the learned value function $V$, among others.

Table 2: Advantage statistics for 20 D4RL tasks.

| Task Name | Positive | Negative | Pos. (%) |
|---|---|---|---|
| halfcheetah-medium | $1.62_{\pm 0.2}$ | $-0.11_{\pm 0.3}$ | 32.8 |
| halfcheetah-medium-replay | $1.84_{\pm 0.3}$ | $-0.90_{\pm 0.0}$ | 30.4 |
| halfcheetah-medium-expert | $2.00_{\pm 0.2}$ | $-1.75_{\pm 0.3}$ | 38.7 |
| hopper-medium | $0.48_{\pm 0.1}$ | $-0.14_{\pm 0.0}$ | 22.1 |
| hopper-medium-replay | $1.25_{\pm 0.2}$ | $-0.83_{\pm 0.1}$ | 38.3 |
| hopper-medium-expert | $0.39_{\pm 0.2}$ | $-0.55_{\pm 0.0}$ | 10.1 |
| walker2d-medium | $0.70_{\pm 0.1}$ | $-0.06_{\pm 0.0}$ | 43.5 |
| walker2d-medium-replay | $1.87_{\pm 0.3}$ | $-0.13_{\pm 0.0}$ | 20.0 |
| walker2d-medium-expert | $2.33_{\pm 0.3}$ | $-2.20_{\pm 0.5}$ | 26.2 |
| antmaze-umaze | $1.70_{\pm 0.2}$ | $-1.05_{\pm 0.1}$ | 52.8 |
| antmaze-umaze-diverse | $0.03_{\pm 0.0}$ | $-0.04_{\pm 0.0}$ | 34.0 |
| antmaze-medium-play | $0.58_{\pm 0.1}$ | $-0.40_{\pm 0.1}$ | 44.7 |
| antmaze-medium-diverse | $0.48_{\pm 0.05}$ | $-0.26_{\pm 0.05}$ | 54.0 |
| antmaze-large-play | $0.58_{\pm 0.02}$ | $-0.28_{\pm 0.05}$ | 48.3 |
| antmaze-large-diverse | $0.42_{\pm 0.08}$ | $-0.23_{\pm 0.02}$ | 62.3 |
| pen-human | $2.34_{\pm 1.1}$ | $-1.74_{\pm 0.6}$ | 41.3 |
| pen-cloned | $1.14_{\pm 0.2}$ | $-1.01_{\pm 0.1}$ | 33.4 |
| kitchen-complete | $1.15_{\pm 0.2}$ | $-0.85_{\pm 0.1}$ | 42.3 |
| kitchen-partial | $0.47_{\pm 0.1}$ | $-0.44_{\pm 0.2}$ | 31.1 |
| kitchen-mixed | $0.52_{\pm 0.0}$ | $-0.74_{\pm 0.0}$ | 33.8 |

**Advantage Soft Clipping.** To prevent unstable learning dynamics caused by extreme advantage values, we apply a soft clipping transformation to all computed advantages. The function is defined as

$$\text{softclip}(x) = \begin{cases} \lambda_p \cdot \tanh\left(\frac{x}{\lambda_p}\right), & x \geq 0, \\ \lambda_n \cdot \tanh\left(\frac{x}{\lambda_n}\right), & x < 0, \end{cases}$$

where $\lambda_p$ and $\lambda_n$ serve as scaling factors for positive and negative values, respectively, replacing the single $\lambda$ used in Eq. (10) to enhance empirical performance in our implementation.

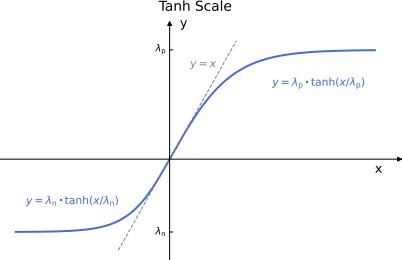

Figure 7: Visualization of the $\text{softclip}(x)$.

This formulation softly bounds the advantage values, with smooth saturation in the tails and a near-linear response around zero. Unlike hard clipping, it avoids sharp discontinuities while preserving the relative differences between actions, which is important for effective Q-function learning. A visualization of the softclip transformation is shown in Figure 7.

In practice, we observe that performance is largely insensitive to the precise values of $\lambda_p$ and $\lambda_n$, as long as their ratio is maintained. Specifically, setting $\lambda_p$ to approximately $1.5 \times \lambda_n$ (e.g., such as $\lambda_p = 6$ and $\lambda_n = 4$) consistently yields stable gradients and expressive advantage signals. These results indicate that the method is robust to the specific choice of these hyperparameters, provided the relative scaling is preserved.

## C.2 DIFFUSION ACTOR AND RESIDUAL-CRITIC ARCHITECTURE DESIGN

Our method jointly optimizes three network modules during main training: a diffusion-based actor, a Q-function critic, and a value function $V$. This section describes the architecture of these components, excluding auxiliary networks used for advantage computation.

**Diffusion Actor.** The actor is instantiated as a denoising diffusion probabilistic model (DDPM) with a variance-preserving (VP) noise schedule. The noise predictor is a five-layer multilayer perceptron

(MLP) with Mish activations. We set the number of denoising steps to 10 across all tasks, balancing expressiveness with computational efficiency.

**Critic Networks.** Both the Q-function and value function are instantiated in two variants: a standard MLP and a residual architecture comprising 16 residual blocks. We observe that the residual networks significantly improve both training stability and final performance in complex tasks, particularly in sparse-reward environments such as AntMaze. We attribute this to the increased representational capacity of the residual architecture, which is better suited to capturing the fine-grained structure of value functions when learning targets are noisy or heterogeneous. In contrast, shallow MLPs tend to underfit in such regimes, leading to unstable or overly conservative estimates. An illustration of the residual block is provided in Figure 8.



Figure 8: Residual block architecture used in critic networks.

### C.3 REPEATED SAMPLING FOR VALUE-GUIDED DIFFUSION POLICIES

Diffusion-based policies are capable of modeling expressive, multimodal action distributions conditioned on state. However, this flexibility introduces sampling stochasticity, where single-sample rollouts may fall into suboptimal modes. To address this, we adopt a repeated sampling strategy guided by the learned Q-function, which enhances both training stability and evaluation performance by enabling more informed action selection.

**Training-Time Sampling: Max-Q Backup.** During critic updates, we widely employ a *Max-Q Backup* mechanism from CQL (Kumar et al., 2020): for each transition, multiple candidate actions are sampled from the policy at the next state, and the Q-target is computed using the maximum or softmax-weighted Q-value among them. This mitigates underestimation bias caused by poor single-sample rollouts and reduces the variance of the TD targets. We observe that modest sample counts (e.g., 3–5) already improve stability, while more complex tasks—such as `halfcheetah-medium-replay` and `antmaze-medium-diverse` benefit from larger sample sizes (e.g., 10). As shown in Figure 9, increasing the number of backup samples leads to higher predicted Q-values and improved empirical returns.

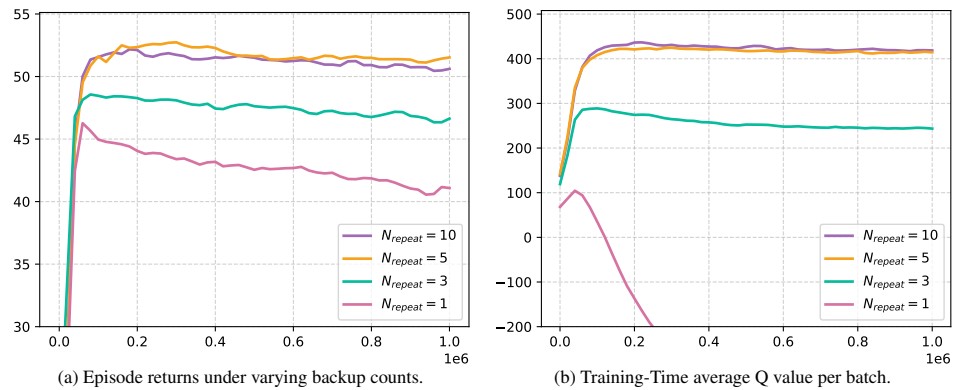

(a) Episode returns under varying backup counts.   (b) Training-Time average Q value per batch.

Figure 9: Impact of repeated sampling in both training and inference stages in `halfcheetah-medium-replay`.

**Evaluation-Time Sampling: Q-Guided Inference.**   At evaluation time, we sample a large set of candidate actions (typically 50–200) from the diffusion model, and select actions via Q-weighted sampling. Specifically, Q-values are transformed into a softmax distribution, from which the final action is drawn. This Q-guided inference biases the policy toward high-value modes while retaining stochasticity. Across tasks, we consistently observe superior returns compared to single-sample decoding.

This effect stems from the multimodal nature of diffusion policies: only a subset of modes yield high return, especially in sparse-reward settings. Without repeated sampling, the critic may overlook these high-reward regions, leading to pessimistic target estimates and suboptimal updates. Expanding the candidate set increases the likelihood of capturing valuable modes, thereby improving both value estimation and policy quality.

## C.4   HYPERPARAMETER SETUP

We highlight several key components of our approach, including the use of the $\kappa$ quantile to define reward thresholds, the integration of residual blocks to enhance the expressiveness of the critic, and repeated sampling to exploit the multimodal capacity of the diffusion model. These components play a central role in achieving strong performance across tasks. Table 3 provides the full set of hyperparameters; all other parameters follow default configurations from DQL (Wang et al., 2022) without further modification.

Table 3: Hyperparameter configurations for all evaluated tasks. We report the quantile threshold $\kappa$, a boolean indicating whether max Q backups are applied, the number of backup times $n$, and the backbone used for the critic network.

| Task | $\kappa$ | Max Q Backup | $n$ | Critic Net |
|---|---|---|---|---|
| halfcheetah-medium | 0.75 | True | 5 | ResNet |
| halfcheetah-medium-replay | 0.75 | True | 5 | ResNet |
| halfcheetah-medium-expert | 0.75 | True | 10 | ResNet |
| hopper-medium | 0.75 | False | 1 | ResNet |
| hopper-medium-replay | 0.75 | True | 5 | ResNet |
| hopper-medium-expert | 0.95 | False | 1 | ResNet |
| walker2d-medium | 0.65 | True | 3 | ResNet |
| walker2d-medium-replay | 0.85 | False | 1 | ResNet |
| walker2d-medium-expert | 0.75 | False | 1 | MLP |
| antmaze-umaze | 0.55 | True | 10 | ResNet |
| antmaze-umaze-diverse | 0.65 | True | 10 | ResNet |
| antmaze-medium-play | 0.65 | True | 10 | ResNet |
| antmaze-medium-diverse | 0.65 | True | 10 | ResNet |
| antmaze-large-play | 0.65 | True | 10 | ResNet |
| antmaze-large-diverse | 0.55 | True | 10 | ResNet |
| pen-human | 0.65 | True | 3 | MLP |
| pen-cloned | 0.65 | True | 3 | MLP |
| kitchen-complete | 0.65 | False | 1 | MLP |
| kitchen-partial | 0.65 | False | 1 | MLP |
| kitchen-mixed | 0.65 | False | 1 | MLP |

## C.5   SENSITIVITY ANALYSIS OF THE PARAMETER $\kappa$

The key innovation of our method lies in the design of the advantage function defined in Eq. (9). This formulation introduces a parameter $\kappa$ to control the level of conservatism in our algorithm. We investigate how $\kappa$ interacts with key factors including dataset quality, task difficulty, and reward sparsity. This investigation aims to provide guidance for tuning ADAC when applying it to new tasks. We conduct ablation studies on six sparse-reward AntMaze environments with three levels of difficulty. We also evaluate on nine Gym Locomotion tasks under three dataset quality settings. As shown in Fig. 10, we vary $\kappa$ over the range $\{0.55, 0.65, 0.75, 0.85, 0.95\}$ to assess its impact.

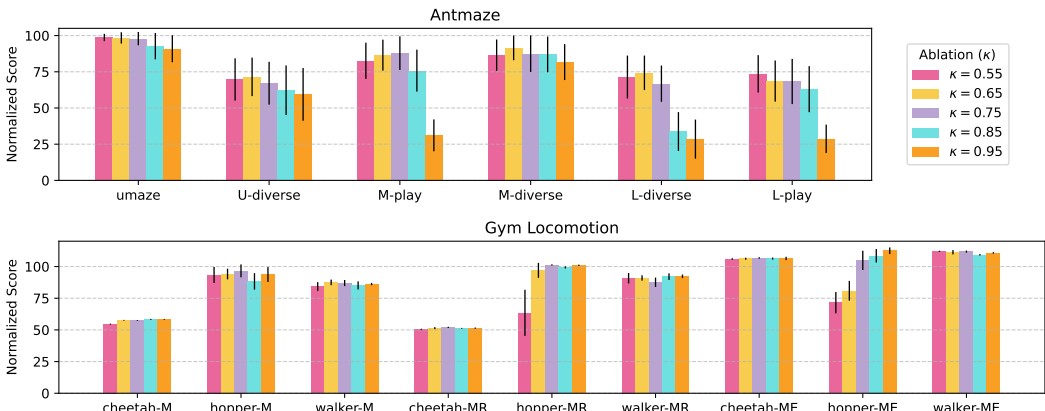

Figure 10: Ablation studies on the effect of $\kappa$, the quantile defining the threshold for positive advantage, in AntMaze and Gym Locomotion domains. Higher values of $\kappa$ (approaching 1) correspond to using the value of the optimal action under the behavior policy as the threshold, while lower $\kappa$ values relax this criterion. All results are averaged over 4 independent random seeds.

We find that in sparse-reward AntMaze tasks, smaller values of $\kappa$ (e.g., 0.55 and 0.65) yield superior performance, whereas in dense-reward Gym tasks, larger values (e.g., 0.75) lead to better results. This observation aligns with the intended design of our method: smaller $\kappa$ values promote the selection of OOD actions, which is essential for achieving high returns in sparse-reward settings.

One potential limitation of our method is the need to sample multiple actions ($N \equiv 25$ in this paper) from the behavior policy to compute the advantage function. However, our ablation study shows that in each task domain, competitive performance can be achieved across at least three different values of $\kappa$, indicating that the algorithm is relatively insensitive to $\kappa$. This suggests that the number of candidate actions can be reduced without significantly affecting performance. We implement our algorithm using the JAX/Flax framework, which offers faster training and inference speed than the DQL method and is comparable to the optimized EDP (Kang et al., 2023) implementation as shown in Figure 6.

## C.6 Auxiliary Model Pretraining

To facilitate offline reinforcement learning, we pretrain two models: a transition model that predicts the next state from a state-action pair, and a behavior cloning model based on a diffusion probabilistic model (DDPM) that learns the action distribution conditioned on the current state. The transition model and the DDPM's noise predictor are both implemented as MLPs with 256 neurons per layer, using the Mish activation. As shown in Table 4, we use a 95%/5% split, batch size 256, and 300 000 gradient steps, optimized with weight-decayed Adam ($3 \times 10^{-4}$ learning rate). Each model trains in under 10 minutes on an NVIDIA RTX 4090. Once trained on a dataset, these models can be reused across experiments, improving efficiency and ensuring consistency regardless of main training hyperparameters.

Table 4: Hyperparameters for Transition and Behavior Cloning Model Pretraining

| Hyperparameter | Transition Model | Behavior Cloning Model |
|---|---|---|
| Architecture | 4-layer MLP 256 neurons per layer | 5-layer MLP 256 neurons per layer |
| Activation Function | Mish | Mish |
| Optimizer | AdamW | AdamW |
| Learning Rate | $3 \times 10^{-4}$ | $3 \times 10^{-4}$ |
| Batch Size | 256 | 256 |
| Gradient Descent Steps | 300 000 | 300 000 |
| Train/Test Split | 95% / 5% | 95% / 5% |

## C.7 DATASET CONSTRUCTION FOR OFFLINE RL IN POINTMAZE

We utilize the `PointMaze` environment from Gymnasium Robotics, a refactored version of D4RL's `Maze2D`. In this environment, an agent navigates a closed maze using 2D continuous force control (bounded $(x, y)$ forces applied at 10 Hz). For our specific experiments focusing on sparse reward spatial navigation, we simplify the state space by omitting goal-related fields. This results in a compact 4-dimensional observation vector comprising only the agent's position and velocity.

To construct the offline dataset, we collect trajectories across multiple runs of online SAC (Haarnoja et al., 2018) training. We employed a staged sampling strategy during online training (Algorithm 2). This strategy involved periodically performing trajectory rollouts using the current policy, allowing us to collect a diverse set of behaviors as the policy evolved throughout training. This collection process spanned 10 independent runs of online SAC training, each for up to 250 000 steps.

We manually collected 853 successful yet sub-optimal trajectories to construct the offline training dataset, each shorter than 1000 steps but significantly longer than the shortest length of approximately 142 steps, which corresponds to a straight-line path from the start to the goal. As shown in the last subfigure of Figure 4, the offline dataset includes no optimal trajectories (length $< 175$), only 3.4% near-optimal (length $< 225$), and 19.1% competitive trajectories (length $< 300$). These trajectories fall into three distinct modes—Left, Middle, and Right routes—depending on which corridor the agent takes to bypass the obstacles (see corresponding subfigures in Figure 4). The dataset the bar plot of Figure 4 comprises 391 391 Q-learning-style tuples of the form $(s, a, s', r, \text{done})$.

---

**Algorithm 2** Trajectory Sampling for Offline Dataset Construction

1: **Initialize:** policy $\pi$, environment $\mathcal{E}$, replay buffer $\mathcal{B}$
2: **Define:** a staged sampling schedule alternating coarse- and fine-grained rollouts
3: **for** each training step **do**
4:     Interact with $\mathcal{E}$ using $\pi$ and store transitions $(s_t, a_t, r_t, s_{t+1})$ into $\mathcal{B}$
5:     Periodically update $\pi$ using mini-batches sampled from $\mathcal{B}$
6:     **if** sampling interval is triggered **then**
7:         Execute coarse-grained or fine-grained trajectory rollout according to schedule
8:         Store resulting trajectories in $\mathcal{D}$
9:     **end if**
10: **end for**
11: Filter suboptimal trajectories based on return and length heuristics
12: **Return:** offline dataset $\mathcal{D}_{\text{offline}}$

---

As detailed in Table 5, the dataset exhibits a distribution heavily skewed towards suboptimal behavior, consistent with our collection strategy. Specifically, it contains no optimal trajectories, a very small number of near-optimal paths (3.4%), and a modest proportion (15.7%) of competitive paths, which we consider to be acceptably successful for the task. The vast majority of the dataset (80.9%) consists of trajectories categorized as Sub-Optimal based on their length, defined as exceeding 300 steps.

Table 5: Distribution of trajectory quality based on length in the collected offline dataset.

| Category | Length Range | Proportion |
|---|---|---|
| Optimal | $< 175$ | 0.0% |
| Near-Optimal | $[175, 225)$ | 3.4% |
| Competitive | $[225, 300)$ | 15.7% |
| Sub-Optimal | $\geq 300$ | 80.9% |

## D DECLARATION

I declare that Large Language Models (LLMs) were used solely for language polishing in this paper. No other usage of LLMs was involved.

