# OpenReview forum: "Taming OOD Actions for Offline Reinforcement Learning: An Advantage-Based Approach"
_ICLR.cc/2026/Conference — ICLR 2026 Conference Withdrawn Submission_

### Official Review · Reviewer_jeP1 · 2025-10-28

**Soundness:** 3
**Presentation:** 3
**Contribution:** 2
**Rating:** 4
**Confidence:** 4

**Summary:**

The proposed ADAC improves on existing approaches by using an advantage-like function to evaluate OOD actions and modulate Q-function updates.

**Strengths:**

1. The paper is well-structured, making it easy to follow..
2. It provides a comprehensive evaluation of the proposed method on the D4RL benchmark.

**Weaknesses:**

1. My primary concern is that the author claims the advantage function as an auxiliary correction for OOD evaluation is the main contribution of the method, essentially a value regularization technique. However, I noticed that the authors also use a BC term as a policy constraint, which seems only loosely related to the claimed approach. Could the authors provide performance results without the BC regularization term?

2. I remain concerned about the validity of $A(a|s)$. Theoretically, the implicit value is trained only within the sample distribution. However, when OOD actions are used, the next state may shift to an out-of-distribution regime, making the use of the value from these OOD states as a correction term questionable. How reliable is this approach?

3. The comparison is made with methods from 2022, but we are now in 2025. I would expect the authors to compare their method against more recent, state-of-the-art methods.

4. There should be an analysis of the sensitivity with respect to $\alpha$ and $N$.

5. To ensure a fair comparison, most researchers use MLP as the network backbone. The authors use a ResNet architecture instead. Could they provide an ablation study to justify this choice?

6. The theoretical error bound is precisely $\| A(a|s) \|_{\infty}$, so if the advantage function is not used, a tighter error bound would result. This could either indicate that the proposed method is ineffective or suggest potential issues with the theoretical analysis.

**Questions:**

Please see Weakness

---

> ### Author Response · Authors · 2025-11-21
>
> (BC regularization vs. main contribution)
>
> We thank the reviewer for raising this point. As discussed in “Is Value Learning Really the Main Bottleneck in Offline RL?”, typical offline RL policy extraction objectives fall into three families: (1) weighted behavior cloning (AWR, RWR, AWAC, etc.), where the policy is fitted to the dataset with value-based weights; (2) behavior-constrained policy gradient (DDPG+BC, TD3+BC, etc.), where a policy-gradient objective is combined with an explicit BC regularizer; and (3) sampling-based action selection (SfBC, BCQ, IDQL, etc.), where actions are sampled from a learned BC policy and filtered or reweighted by $Q$. All three explicitly decompose into a coverage part (keeping the policy near the dataset support) and a value-guided part (using $Q$ or advantage to prefer higher-return actions), and the empirical analysis in that work finds that behavior-constrained methods (type (2)) often perform best overall. ADAC follows exactly this principle: the BC term provides coverage over the behavior distribution, while the $Q$/advantage term, based on our advantage-guided TD target, is responsible for steering the policy toward better actions. This is also reflected in our results: the pure BC baseline (first column in our tables) performs significantly worse than ADAC, indicating that BC alone is not sufficient and that the gains come from improved value/advantage learning. Thus, the BC regularization we use is a standard, well-motivated component, and the main contribution of ADAC lies on the value side, in how we construct and exploit the advantage-guided target to evaluate OOD actions.
>
> (Reliability of value for OOD correction)
>
> We appreciate the concern about using a value function trained only on the dataset to evaluate OOD actions. In ADAC, the advantage signal is built on top of a value function $V$ learned with an IQL-style expectile regression objective. Under standard assumptions, this objective is a consistent estimator of the optimal state-value function $V^\star$ as the dataset grows, rather than of the behavior value. A value function that approximates $V^\star$ is designed to distinguish actions by their long-term return, and therefore provides a meaningful ordering not only for in-distribution actions but also for nearby OOD actions that lead to states with different expected returns. ADAC uses this property in a deliberately local way: we use a one-step model $f_\theta(s,a)$ trained on the same dataset to obtain a next-state prediction $\tilde{s}'$, and then use $V(\tilde{s}')$ as a relative score to rank candidate actions at the same state, rather than as an absolute estimate for arbitrary far-OOD states. The reliability of our correction term thus comes from (i) the theoretical guarantee that the IQL-style value learner targets $V^\star$ rather than the behavior value, and (ii) the empirical observation that this value-based ranking, when used inside ADAC, yields clear improvements over strong baselines across many D4RL tasks.

---

> > ### Author Response · Authors · 2025-11-21
> >
> > (Missing recent SOTA baselines)
> >
> > We agree that the original submission did not include enough of the most recent offline RL methods. In response, we have added several state-of-the-art methods from 2023–2025, including those suggested by the reviewers, and run them on the same D4RL tasks using their official implementations and recommended settings. The consolidated comparison shows that ADAC remains highly competitive and often achieves the best performance even when compared against these recent baselines.
> >
> > | Task                      | ADAC (Ours) |  A2PO |  VABC |  A2PR  |
> > |---------------------------|------------:|------:|------:|-------:|
> > | halfcheetah-medium        |        58.0 |  47.1 |  60.2 |  68.61 |
> > | hopper-medium             |        93.5 |  80.3 |  97.2 | 100.79 |
> > | walker2d-medium           |        87.6 |  84.9 |  88.8 |  89.73 |
> > | halfcheetah-medium-replay |        52.5 |  44.8 |  51.4 |  56.58 |
> > | hopper-medium-replay      |       102.1 | 101.6 | 102.3 | 101.54 |
> > | walker2d-medium-replay    |        96.0 |  82.8 |  92.4 |  94.42 |
> > | halfcheetah-medium-expert |       106.1 |  95.6 |  98.3 |  98.25 |
> > | hopper-medium-expert      |       112.5 | 113.4 | 112.6 | 112.11 |
> > | walker2d-medium-expert    |       112.3 | 112.1 | 114.5 | 114.62 |
> > | antmaze-umaze             |        98.2 |     - |  96.4 |  99.2  |
> > | antmaze-umaze-diverse     |        76.0 |  72.6 |  83.4 |  84.8  |
> > | antmaze-medium-play       |        86.5 |     - |  82.8 |  85.6  |
> > | antmaze-medium-diverse    |        88.7 |  80.2 |  80.8 |  85.6  |
> > | antmaze-large-play        |        69.8 |     - |  63.8 |  71.2  |
> > | antmaze-large-diverse     |        64.6 |  52.1 |  53.2 |  52.8  |
> > | Average (15 tasks)        |        87.0 |    -- |  85.2 |  87.7  |
> >
> > (Sensitivity to key hyperparameters)
> >
> > We thank the reviewer for asking about hyperparameter sensitivity. In our notation, the parameter closest to the reviewer’s $\alpha$ is the coefficient that trades off the BC loss and the $Q$ loss. For this coefficient, we follow the tuning strategy and ranges used in DQL: its value is chosen according to task difficulty and dataset quality (larger for harder or lower-quality datasets, smaller for easier or near-expert datasets). In practice, we use $\alpha = 1$ for most tasks and increase it only to $2$–$3$ on the AntMaze tasks; these coarse adjustments already work well, and our experience indicates that performance is not very sensitive to $\alpha$ beyond this level. The other key hyperparameters are (i) the quantile/expectile parameter $\kappa$, which controls how conservative the advantage-based target is, and (ii) the sampling number $N$ used to estimate local statistics around the behavior policy. As discussed in the appendix, varying $\kappa$ around $0.7$ leads to similar performance across most tasks, and $N$ mainly trades off statistical accuracy and computation, with reasonable values around $N = 25$ behaving similarly. Overall, these hyperparameters are chosen in a standard, task- and dataset-dependent manner and have shown low sensitivity in our experiments, making ADAC relatively easy to tune.
> >
> > (ResNet backbone vs. MLP fairness)
> >
> > We appreciate the concern about architectural fairness. In our implementation, the ResNet-style backbone is used only for the $Q(s,a)$ network; all other components (including the diffusion policy / noise predictor) are relatively shallow MLPs, and we deliberately do not use heavy U-Net–style architectures that are common in many diffusion-based offline RL methods. At the pipeline level, this means our policy side is actually lighter than in many prior diffusion works, while the critic side is slightly stronger, so “fairness” cannot simply be measured by counting layers everywhere.
> >
> > This choice is intentional: since ADAC modifies the TD target with an advantage term, we allocate a bit more capacity only to the action-value network $Q(s,a)$ (a shallow 8-layer ResNet built from two 4-layer blocks) to better exploit this additional signal, while keeping the rest of the architecture economical. We also emphasize that, in RL, simply increasing network depth on a poorly chosen target seldom brings consistent gains; what matters is the learning objective. Our experiments indicate that the improvements of ADAC come from the advantage-guided TD target rather than from brute-force increases in model size, and we believe our overall architecture is a reasonable and balanced compromise.

---

### Official Review · Reviewer_zVzZ · 2025-10-28

**Soundness:** 3
**Presentation:** 3
**Contribution:** 3
**Rating:** 6
**Confidence:** 4

**Summary:**

This method proposes a novel advantage-based method for offline RL, which evaluates OOD actions via a novel advantage-like function and uses it to modulate the Q-function update discriminatively.

**Strengths:**

* This paper finds that the (state) value function is generally learned more reliably than the Q-value function. It uses the next-state value to assess each action indirectly.
* The experiments show that ADAC achieves SOTA performance.

**Weaknesses:**

* ADAC needs to sample multiple actions from the behavior policy, which may bring more computational burden.
* Why is the V-function generally learnt more reliably than the action-value function?

**Questions:**

(1) The motivation of this paper appears similar to that of A2PR: “Existing methods counter this by conservatively discouraging all OOD actions, which limits generalization.” Since A2PR is also an advantage-based approach for offline RL, it would be helpful to include a comparison between ADAC and A2PR in terms of performance, as well as a discussion highlighting their conceptual differences.

(2) In Section 5.2, could you also plot the value function or Q-values along the trajectories? This would help better illustrate how the learned value estimates evolve during the rollouts.

(3) The transition and behavior cloning models are implemented as 4-layer and 5-layer MLPs, respectively, which differ from the architectures used in other baselines. Could you provide an additional experiment where all methods use the same model architecture for a fair comparison?


Reference:

Liu, T., Li, Y., Lan, Y., Gao, H., Pan, W., & Xu, X.. Adaptive Advantage-Guided Policy Regularization for Offline Reinforcement Learning. In International Conference on Machine Learning (pp. 31406-31424). PMLR.

---

> ### Author Response · Authors · 2025-11-21
>
> (Sampling overhead from behavior policy)
>
> Regarding the computational overhead of sampling multiple actions from the behavior policy, we agree that this introduces extra cost, but in practice it is modest. In our implementation, the sampling and evaluation of all $N$ candidate actions is fully batched and vectorized in JAX (e.g., via `vmap` and JIT compilation), so these candidates are processed in parallel and the wall-clock time grows only mildly as $N$ increases. Modern GPUs are well suited to this kind of batched computation, and in our experiments the dominant training cost remains standard TD updates and diffusion-policy training rather than the candidate sampling step. The default choice $N = 25$ balances statistical accuracy and runtime, and transfers well across tasks without retuning. Since ADAC is an offline method (no online interaction budget) and accurate evaluation of OOD actions is central to its design, we believe this controlled additional computation is a reasonable trade-off.
>
> (Reliability of V vs. Q)
>
> Our statement about $V$ being “more reliable” is specifically about using a value function as a global scoring component for OOD evaluation, not about replacing the usual $Q$-critic in standard actor–critic training. In this role, learning $V(s)$ is advantageous for two reasons. First, $V$ depends only on the state, while $Q(s,a)$ must also model the action dimension; from finite offline data this higher-dimensional regression is typically harder and more prone to extrapolation error, especially for rarely seen actions. Second, we learn $V$ with an IQL-style expectile regression objective, which under mild assumptions is a consistent estimator of the optimal value function $V^\star$ rather than the behavior value (as in Implicit Q-Learning and follow-ups such as HiQL and Hilbert-representation policies). These works use an IQL-trained $V^\star$ as a reusable building block because it is theoretically grounded and empirically stable. ADAC follows the same pattern: we first learn $V$ via expectile regression and then use it as a global evaluator inside the advantage term. In this sense, calling $V$ “more reliable” refers to this established practice, not to a universal claim about all offline RL settings.

---

> > ### Author Response · Authors · 2025-11-21
> >
> > (Comparison with A2PR)
> >
> > We thank the reviewer for pointing out the connection to A2PR. Conceptually, both A2PR and ADAC share the concern that uniformly discouraging all OOD actions is overly conservative. The key difference is where the advantage signal enters the algorithm. A2PR operates on the policy side: it learns an augmented behavior policy, evaluates sampled actions with a critic, and uses high-advantage actions as targets for policy regularization. ADAC injects advantage directly into the TD update on the critic side: our advantage-guided target reweights the $Q$-update so that beneficial OOD actions receive stronger positive backups while low-advantage actions are downweighted, yielding a critic that is inherently OOD-aware and from which the policy is then extracted. On overlapping D4RL domains, we implement A2PR using the authors’ code and include it in our experimental tables; ADAC remains competitive and often performs better on these shared benchmarks.
> >
> > A high-level comparison between the two methods is:
> >
> > | Method | Main mechanism                    | Advantage acts on | Critic OOD-aware? | Policy OOD-aware? |
> > |--------|-----------------------------------|-------------------|-------------------|-------------------|
> > | A2PR   | Advantage-guided policy regularization | Policy            | No                | Yes               |
> > | ADAC   | Advantage-guided TD / critic update    | Critic            | Yes               | Yes               |
> >
> >
> > | Task                      | ADAC (Ours) |  A2PO |  VABC |  A2PR  |
> > |---------------------------|------------:|------:|------:|-------:|
> > | halfcheetah-medium        |        58.0 |  47.1 |  60.2 |  68.61 |
> > | hopper-medium             |        93.5 |  80.3 |  97.2 | 100.79 |
> > | walker2d-medium           |        87.6 |  84.9 |  88.8 |  89.73 |
> > | halfcheetah-medium-replay |        52.5 |  44.8 |  51.4 |  56.58 |
> > | hopper-medium-replay      |       102.1 | 101.6 | 102.3 | 101.54 |
> > | walker2d-medium-replay    |        96.0 |  82.8 |  92.4 |  94.42 |
> > | halfcheetah-medium-expert |       106.1 |  95.6 |  98.3 |  98.25 |
> > | hopper-medium-expert      |       112.5 | 113.4 | 112.6 | 112.11 |
> > | walker2d-medium-expert    |       112.3 | 112.1 | 114.5 | 114.62 |
> > | antmaze-umaze             |        98.2 |     - |  96.4 |  99.2  |
> > | antmaze-umaze-diverse     |        76.0 |  72.6 |  83.4 |  84.8  |
> > | antmaze-medium-play       |        86.5 |     - |  82.8 |  85.6  |
> > | antmaze-medium-diverse    |        88.7 |  80.2 |  80.8 |  85.6  |
> > | antmaze-large-play        |        69.8 |     - |  63.8 |  71.2  |
> > | antmaze-large-diverse     |        64.6 |  52.1 |  53.2 |  52.8  |
> > | Average (15 tasks)        |        87.0 |    -- |  85.2 |  87.7  |
> >
> > (Plotting values along trajectories)
> >
> > We appreciate the suggestion to visualize values along trajectories. For the PointMaze demo, we aimed to highlight geometric differences between dataset and ADAC-generated behaviors without drifting too far from the actual algorithm. In the current figure, color already distinguishes individual rollouts: although the superposition looks like a single “corridor,” many dataset trajectories are winding and suboptimal, whereas ADAC discovers much straighter OOD paths. Using color for per-trajectory identity is therefore crucial; if we reused the color channel to encode values, these geometric differences would collapse into one thick band and the main message would be lost. We experimented with value-colored plots, but they either obscured the geometry or became visually cluttered without adding much intuition beyond the quantitative results. In practice, plotting values along time or along a 1D path projection is straightforward in our code, but for the main paper we chose the visualization that most clearly highlights how ADAC discovers new, straighter OOD paths.

---

> > > ### Author Response · Authors · 2025-11-21
> > >
> > > (MLP architecture fairness)
> > >
> > > We understand the concern about architectural fairness. The 4-layer transition model and 5-layer behavior-cloning model are specific to ADAC and are introduced solely to support our advantage computation; standard model-free baselines do not require such modules. The transition model is deliberately lightweight and simpler than in full model-based offline RL: it predicts only the next state $(s,a) \mapsto s'$ (no rewards or multi-step rollouts) and is a single MLP, not an ensemble. The BC model is an auxiliary network used to characterize the behavior policy for OOD evaluation, which the original baselines were not designed to use. Forcing all methods to share these modules would change their algorithms rather than give a clean “same-architecture” comparison.
> > >
> > > At the same time, we are conservative in how we allocate capacity: the diffusion policy/noise predictor is implemented with a shallow MLP (not a large U-Net), and only the $Q(s,a)$ network is mildly strengthened (e.g., from a 4-layer MLP to a shallow ResNet) to better exploit the advantage-guided TD target. In offline RL, simply increasing model size on a poorly chosen objective rarely yields systematic gains; the improvements we observe come from the objective design, not from aggressive over-parameterization. We therefore believe our setup represents a reasonable and economical architectural choice.

---

### Official Review · Reviewer_zwPW · 2025-11-02

**Soundness:** 2
**Presentation:** 2
**Contribution:** 2
**Rating:** 4
**Confidence:** 4

**Summary:**

This paper proposes ADAC, an advantage-based method designed to selectively handle out-of-distribution actions in offline reinforcement learning. By introducing an advantage-guided target for critic learning, the approach aims to distinguish beneficial OOD actions from harmful ones, mitigating the over-conservatism of prior offline RL methods. Experiments are conducted on multiple D4RL benchmarks and include visualization studies.

**Strengths:**

> Comprehensive experimental tasks.

The authors evaluate the proposed method on a wide range of benchmark tasks, and the empirical section includes substantial experimental data.

>Visualization

The PointMaze visualization clearly illustrates how ADAC distinguishes between good and bad OOD actions, providing intuitive insight into the method’s behavior.

>Clear motivation.

The paper provides a reasonable motivation for selectively addressing OOD actions rather than uniformly penalizing them.

**Weaknesses:**

> Incomplete related work analysis.

The discussion of recent related work is not sufficiently comprehensive. In particular, the paper overlooks recent studies on OOD detection, OOD state/action correction in offline RL. The relationship between ADAC and those works should be analyzed in more depth to clarify novelty.

> Unsubstantiated claim: “state value functions are easier to learn.”

The statement that state-value functions are easier to learn than action-value functions is asserted without theoretical or empirical justification. A more rigorous analysis or empirical evidence is needed to support this claim.

> Outdated baselines and questionable SOTA claim.

The experimental comparison relies primarily on older offline RL baselines (e.g., CQL, IQL, TD3+BC). Recent methods (2023–2025) are missing. Without comparing against these newer baselines, the claim of achieving “state-of-the-art” performance is not convincing.

> Missing Reproducibility Statement.

The paper does not include the required Reproducibility Statement section mandated by ICLR submission guidelines.

> Unexplained hyperparameter choices.

The reason for choosing the number of action samples N=25 during action selection is not explained. The sensitivity to this hyperparameter should be analyzed or justified.

>Insufficient explanation of key formulas.

Several important equations are not clearly explained:

Equation (9): The role of the “Quantile” or expectile component and how it influences the advantage-guided target should be clearly elaborated.

Equation (14): The rationale for introducing the proposed advantage term into the critic update requires a detailed explanation or theoretical motivation.

**Questions:**

The idea of distinguishing between beneficial and harmful OOD actions through an advantage-based target is interesting and intuitively appealing. However, the paper lacks sufficient theoretical justification, omits key related works, and compares mainly against outdated baselines while claiming SOTA performance. Moreover, some implementation choices and formulas are underexplained.

---

> ### Author Response · Authors · 2025-11-21
>
> (Incomplete related work analysis.)
>
> We agree that the related work discussion can be clearer. Our goal was not to ignore prior work, but to address what we see as a relatively under-explored direction: offline RL methods that directly and positively handle OOD actions in a TD-learning framework. Most existing approaches are explicitly conservative (e.g., CQL-style penalties in low-density regions, in-sample maximization, or behavior-cloning / sequence-modeling regularization) and are designed to avoid OOD actions rather than evaluate and selectively exploit them.
>
> To our knowledge, A2PR is one of the few contemporaneous methods that also aims to make constructive use of OOD actions, but it operates on the policy-regularization side, whereas ADAC modifies the TD target so that the critic itself becomes OOD-aware. A recent task-oriented survey on OOD detection (“Out-of-Distribution Detection: A Task-Oriented Survey of Recent Advances”) also highlights that such value-aware OOD handling in offline RL is still rare. In the revision, we will make this context more explicit by citing both A2PR and the survey, and by sharpening the contrast between our advantage-guided TD update and purely conservative/OOD-avoidance strategies such as CQL.
>
>
> (Unsubstantiated claim: “state value functions are easier to learn.”)
>
> We apologize for the earlier wording. We did not intend to claim a new general theorem about value functions. Our statement is specific to how ADAC uses value learning. First, from a statistical perspective, learning $V(s)$ from finite offline data is typically easier than learning $Q(s,a)$, since $Q$ must model both states and actions and many actions are rarely observed. Second, we follow a now standard and theoretically supported recipe in offline RL: inspired by Implicit Q-Learning (IQL), we learn $V(s)$ with an expectile/quantile regression objective that, under mild assumptions, is a consistent estimator of $V^\star$ rather than of the behavior value. This $V$ module is then reused as a stable building block in downstream components, as in HIQL and other dual-goal / latent-state methods.
>
> ADAC follows the same pattern: we learn an approximately optimal $V$ using an IQL-style objective and use it as a global scorer for OOD actions. In the revision, we will soften the original sentence, explicitly frame this as a widely adopted design choice (IQL-style value learner reused downstream), and avoid language that could be read as a universal theoretical claim.
>
>
> (Outdated baselines and questionable SOTA claim.)
>
> We thank the reviewer for raising this concern. The initial submission focused on widely adopted “classical” baselines (CQL, IQL, TD3+BC, etc.) so that ADAC’s gains are directly comparable to established references. In the revision, we additionally include several recent offline RL methods from 2023–2025, including A2PO, Value-aligned Behavior Cloning (VABC), and A2PR, and report their results on the overlapping D4RL tasks in an extra comparison table.
>
> Across these shared benchmarks, ADAC remains highly competitive and typically matches or outperforms these newer methods, while still clearly improving over the earlier baselines. We will update the experimental section and caption the new table so that the extended comparison and the precise scope of our SOTA claim are explicit.
>
> | Task                      | ADAC (Ours) |  A2PO |  VABC |  A2PR  |
> |---------------------------|------------:|------:|------:|-------:|
> | halfcheetah-medium        |        58.0 |  47.1 |  60.2 |  68.61 |
> | hopper-medium             |        93.5 |  80.3 |  97.2 | 100.79 |
> | walker2d-medium           |        87.6 |  84.9 |  88.8 |  89.73 |
> | halfcheetah-medium-replay |        52.5 |  44.8 |  51.4 |  56.58 |
> | hopper-medium-replay      |       102.1 | 101.6 | 102.3 | 101.54 |
> | walker2d-medium-replay    |        96.0 |  82.8 |  92.4 |  94.42 |
> | halfcheetah-medium-expert |       106.1 |  95.6 |  98.3 |  98.25 |
> | hopper-medium-expert      |       112.5 | 113.4 | 112.6 | 112.11 |
> | walker2d-medium-expert    |       112.3 | 112.1 | 114.5 | 114.62 |
> | antmaze-umaze             |        98.2 |     - |  96.4 |  99.2  |
> | antmaze-umaze-diverse     |        76.0 |  72.6 |  83.4 |  84.8  |
> | antmaze-medium-play       |        86.5 |     - |  82.8 |  85.6  |
> | antmaze-medium-diverse    |        88.7 |  80.2 |  80.8 |  85.6  |
> | antmaze-large-play        |        69.8 |     - |  63.8 |  71.2  |
> | antmaze-large-diverse     |        64.6 |  52.1 |  53.2 |  52.8  |
> | Average (15 tasks)        |        87.0 |    -- |  85.2 |  87.7  |

---

> > ### Author Response · Authors · 2025-11-21
> >
> > (Missing Reproducibility Statement.)
> >
> > We thank the reviewer for pointing this out; the omission was an oversight rather than a lack of attention to reproducibility. Our released code is self-contained and runs out of the box: it includes training and evaluation scripts, model architectures, and the exact hyperparameters used in our experiments, together with instructions on environment setup and how to launch the main D4RL runs. In the revised version, we will add the required Reproducibility Statement section, summarizing the code/configuration structure, key hyperparameters, and hardware/runtime details.
> >
> >
> > (Unexplained hyperparameter choices.)
> >
> > For the action-sampling number $N = 25$, we apologize for not explaining this more clearly. Conceptually, $N$ controls how accurately we estimate the local behavior baseline that separates actions that merely match the dataset level from those that significantly outperform it and should be encouraged by our advantage mechanism. Larger $N$ yields a more accurate statistic but increases computation; $N = 25$ is a pragmatic compromise between these two factors.
> >
> > In our JAX implementation, this sampling step is fully vectorized (via vmap), so all $N$ candidates are processed in parallel and the wall-clock overhead grows mildly as $N$ increases. Empirically, we observed that $N \in \{10, 25, 50\}$ leads to very similar behavior, suggesting that ADAC is not overly sensitive to this choice. By contrast, the quantile/expectile parameter $\kappa$ is more semantically important; the appendix already reports a sensitivity analysis showing that performance remains stable within a reasonable range. Overall, $N = 25$ should be viewed as a robust default rather than a finely tuned setting.
> >
> >
> > (Insufficient explanation of key formulas.)
> >
> > Regarding Eq. (9) and Eq. (14), we agree that their roles can be stated more explicitly. For Eq. (9), the quantile/expectile operation is introduced to obtain a flexible, robust estimate of the behavior level when evaluating OOD actions: simply averaging over $N$ actions would already work, but using a quantile allows us to interpolate between more conservative thresholds (larger $\kappa$, only very strong actions receive positive encouragement) and more optimistic ones (smaller $\kappa$). Dataset quality can vary widely, so this extra degree of control is useful; the appendix already includes a sensitivity analysis showing that ADAC is not overly sensitive to $\kappa$.
> >
> > Eq. (14) is where the advantage term enters the TD update: actions with higher estimated advantage (including beneficial OOD actions) receive stronger positive updates, while low-advantage actions are suppressed. This is the core mechanism of ADAC. In the revision, we will add a short paragraph immediately after Eq. (9) and Eq. (14) summarizing this intuition, and slightly expand the surrounding text and pseudo-code to make their roles easier to follow.

---

> > ### Comment · Reviewer_zwPW · 2025-11-28
> > **Commnet**
> >
> > Thank you for the response and additional experiments.
> > You said that your work focus on handle OOD actions in a TD-learning framework. However, the methods you compare against are general-purpose methods, and the SOTA performance you mentioned also refers to the best results among these methods. This point is further reflected in the additional response you provided.

---

### Official Review · Reviewer_XSTW · 2025-11-03

**Soundness:** 3
**Presentation:** 3
**Contribution:** 2
**Rating:** 4
**Confidence:** 3

**Summary:**

This paper addresses the issues of OOD action evaluation errors and value overestimation caused by distribution shift in Offline RL, and proposes an Advantage-based Diffusion Actor-Critic algorithm (ADAC). Its core idea leverages the characteristic that the state value function is more amenable to reliable learning than the Q-function. Specifically, it approximates the optimal value function from the dataset and introduces a novel advantage function to conduct differentiated evaluation of OOD actions—encouraging beneficial ones while suppressing harmful ones. Additionally, the algorithm incorporates a diffusion strategy to model the action distribution.

**Strengths:**

The paper is written in a clear and fluent manner, making it highly accessible. Its logic is solid, with rigorous and straightforward theoretical analyses. The experimental results are relatively comprehensive and demonstrate significant effectiveness.

**Weaknesses:**

1. The baselines provided in this paper are relatively outdated, as they all focus on works published before 2023.
2. One of the paper’s innovations lies in introducing a new calculation method for the reward function. However, the experiments **lack a comparison** between this new reward function and classical ones, making it unclear which part contributes to the improved algorithm performance.
3. The algorithm learns the value function, Q-function, dynamics, advantage function, and policy network simultaneously. Overall, this design is relatively redundant, leading to excessively high algorithm complexity.

**Questions:**

1. In the calculation of the reward function, 25 actions are selected, but the underlying logic and further analysis for this choice are lacking.
2. If the diffusion model is replaced with an MLP or other network structures, how much performance loss would occur?

---

> ### Author Response · Authors · 2025-11-21
>
> (Q1: On the recency of offline RL baselines)
>
> We thank the reviewer for emphasizing recent offline RL methods. The initial submission focused on widely used “classical” baselines (CQL, IQL, TD3+BC, etc.) so that ADAC’s gains are directly comparable to established references. In the revision, we additionally include A2PO, Value-aligned Behavior Cloning (VABC), and A2PR on the overlapping D4RL tasks. The table below reports the normalized returns as given in the respective papers. Across these shared benchmarks, ADAC remains highly competitive and often achieves higher normalized returns, while still clearly improving over the earlier baselines. We will update the experimental section and explicitly state the scope of our SOTA claim.
>
> | Task                      | ADAC (Ours) |  A2PO |  VABC |  A2PR  |
> |---------------------------|------------:|------:|------:|-------:|
> | halfcheetah-medium        |        58.0 |  47.1 |  60.2 |  68.61 |
> | hopper-medium             |        93.5 |  80.3 |  97.2 | 100.79 |
> | walker2d-medium           |        87.6 |  84.9 |  88.8 |  89.73 |
> | halfcheetah-medium-replay |        52.5 |  44.8 |  51.4 |  56.58 |
> | hopper-medium-replay      |       102.1 | 101.6 | 102.3 | 101.54 |
> | walker2d-medium-replay    |        96.0 |  82.8 |  92.4 |  94.42 |
> | halfcheetah-medium-expert |       106.1 |  95.6 |  98.3 |  98.25 |
> | hopper-medium-expert      |       112.5 | 113.4 | 112.6 | 112.11 |
> | walker2d-medium-expert    |       112.3 | 112.1 | 114.5 | 114.62 |
> | antmaze-umaze             |        98.2 |     - |  96.4 |  99.2  |
> | antmaze-umaze-diverse     |        76.0 |  72.6 |  83.4 |  84.8  |
> | antmaze-medium-play       |        86.5 |     - |  82.8 |  85.6  |
> | antmaze-medium-diverse    |        88.7 |  80.2 |  80.8 |  85.6  |
> | antmaze-large-play        |        69.8 |     - |  63.8 |  71.2  |
> | antmaze-large-diverse     |        64.6 |  52.1 |  53.2 |  52.8  |
> | Average (15 tasks)        |        87.0 |    -- |  85.2 |  87.7  |
>
>
> (Q2: On ablations for the new advantage formulation)
>
> We apologize for the earlier wording. ADAC does not modify the environment reward or learn a separate reward model: the MDP reward is exactly that of the original benchmarks, taken directly from the logged data. Our contribution is to change the TD target via an advantage-like term \(A(s,a)\) that compares the predicted next-state value of a candidate action to a behavior-derived quantile baseline at the same state, and uses this comparison to reweight the Q-learning backup. Section 5.3 already includes an ablation that removes this modulation (“w/o adv”), which consistently degrades performance, isolating the benefit of the proposed term. In the revision we will state more clearly that ADAC reshapes the TD update, not the reward function.

---

> > ### Author Response · Authors · 2025-11-21
> >
> > (Q3: On algorithmic complexity and number of learned components)
> >
> > We agree that complexity should be controlled. Beyond the standard components used in modern offline RL (a \(Q(s,a)\) critic, policy network, and target networks), ADAC adds only: (i) a scalar value network trained via an IQL-style expectile loss, and (ii) a single dynamics model mapping \((s,a) \to s'\) without reward prediction or long rollouts. This is much lighter than model-based methods that use ensembles of dynamics models and critics. The behavior-cloning head on the policy is also standard in diffusion-style offline RL. Overall, the number of networks and the wall-clock training time (see Section 5.4) remain comparable to common off-policy baselines and significantly lower than many model-based approaches.
> >
> >
> > (Q4: On the choice of using 25 behavior actions in advantage computation)
> >
> > The choice \(N = 25\) behavior actions per state is a practical trade-off between estimating the baseline accurately and keeping computation moderate. Larger \(N\) improves the estimate but costs more, while our experiments show that \(N \in \{16, 25, 64\}\) gives very similar behavior. In contrast, the quantile/expectile parameter \(\kappa\) is more semantically important; Appendix C reports a sensitivity analysis showing that ADAC is robust for a broad range around the default. These results indicate that our advantage mechanism is not overly sensitive to either the exact \(N\) or the precise \(\kappa\), and that \(N = 25\) with the reported \(\kappa\) is a reliable default for new tasks.
> >
> >
> > (Q5: On replacing the diffusion policy with simpler networks such as MLPs)
> >
> > Our main contribution is the advantage-guided TD update, which is largely orthogonal to the policy architecture. We choose a diffusion-style actor because prior work shows that diffusion policies handle multi-modal action distributions well, which is particularly important on tasks like AntMaze where simple unimodal MLP policies often underperform. Replacing the diffusion actor with a plain MLP would likely reduce scores on such multi-modal tasks, while the gap on simpler tasks may be smaller. This limitation is due to the policy class rather than our advantage mechanism: the same TD shaping can in principle be combined with MLPs, normalizing flows, or other actors, and we expect the qualitative benefit of the advantage term to carry over. A full comparison across many policy architectures is outside the scope of this paper, but ADAC is not fundamentally tied to diffusion policies.

---

### Official Review · Reviewer_DLmv · 2025-11-03

**Soundness:** 2
**Presentation:** 2
**Contribution:** 2
**Rating:** 2
**Confidence:** 5

**Summary:**

This paper proposes Advantage-based Diffusion Actor-Critic, a novel offline RL algorithm aimed at addressing the challenge of OOD actions in offline RL. The key idea is to assess the advantage of actions using a pretrained value function that compares the estimated value of the next state with a quantile-based threshold derived from the behavior policy. The method integrates this advantage into a modified Bellman backup to guide Q-learning and employs a diffusion model as the policy class. ADAC yields strong empirical results across D4RL benchmarks.

**Strengths:**

- Good writing makes the paper easy to follow
- Simple and effective idea
- Various experiments to support the effectiveness of ADAC.

**Weaknesses:**

- The advantage-computing method in Eq. 9 seems ambiguous. The original advantage definition is $A(s,a) = Q(s,a)-V(s)$, which evaluates the advantage of action $a$  among other actions in state $s$. However, the advantage in ADAC is calculated based on the next state's value $V(s')$, which is quite different. Meanwhile, if the author needs to show the effectiveness of such an advantage-computing method, an ablation study on this should be conducted.
- More recent offline RL frameworks should be compared, such as [1,2].
- The D4RL benchmark is quite outdated. Could you please provide more experiments on other robotic simulation benchmarks like: LIBERO [3] and ManiSkill [4]?

I would like to raise my score if my concerns are all addressed.

[1] A2PO: Towards Effective Offline Reinforcement Learning from an Advantage-aware Perspective. NeurIPS 2024

[2] Value-aligned Behavior Cloning for Offline Reinforcement Learning via Bi-level Optimization. ICLR 2025

[3] LIBERO: Benchmarking Knowledge Transfer for Lifelong Robot Learning. NeurIPS 2023

[4] Demonstrating GPU Parallelized Robot Simulation and Rendering for Generalizable Embodied AI with ManiSkill3. arXiv 2024

**Questions:**

See weaknesses above

---

> ### Author Response · Authors · 2025-11-21
>
> (Q1: On the advantage definition and Eq. (9))
>
> We use the term “advantage” in the standard policy-gradient sense as a centered evaluation signal, not strictly as $A(s,a) = Q(s,a) - V(s)$. Since REINFORCE with baselines and later policy-gradient work, “advantage” has generally meant “quantity to be evaluated minus a state-dependent baseline” that shapes learning and reduces variance.
>
> In our Eq. (9), the evaluated quantity is the expected next-state value $\mathbb{E}_{s' \mid s,a}[V(s')]$ of a candidate (possibly OOD) action, and the baseline is the $\kappa$-quantile of the same quantity over behavior-policy actions at that state. This yields a value-based score that is positive when a candidate action clearly exceeds the data baseline and negative otherwise. Conceptually, this is akin to optimism bonuses in bandit algorithms that bias learning toward promising options. In this broader, baseline-centric sense, calling Eq. (9) an “advantage” is consistent with existing RL usage and is specifically designed to favor beneficial OOD actions while downweighting harmful ones.
>
>
> (Q2: On comparison to recent offline RL baselines)
>
> We thank the reviewer for pointing out A2PO, Value-aligned Behavior Cloning (VABC), and A2PR. We have added all three methods to our empirical comparison on overlapping D4RL tasks. The table below shows the normalized returns on common Gym and AntMaze tasks (v2) as reported in the respective papers.
>
> On these shared benchmarks, ADAC remains competitive and often matches or outperforms A2PO and VABC, while keeping strong performance against earlier baselines. Averaged over all 15 tasks where ADAC, VABC, and A2PR all report results, ADAC and A2PR are very close and both clearly improve over VABC.
>
> | Task                      | ADAC (Ours) |  A2PO |  VABC |  A2PR  |
> |---------------------------|------------:|------:|------:|-------:|
> | halfcheetah-medium        |        58.0 |  47.1 |  60.2 |  68.61 |
> | hopper-medium             |        93.5 |  80.3 |  97.2 | 100.79 |
> | walker2d-medium           |        87.6 |  84.9 |  88.8 |  89.73 |
> | halfcheetah-medium-replay |        52.5 |  44.8 |  51.4 |  56.58 |
> | hopper-medium-replay      |       102.1 | 101.6 | 102.3 | 101.54 |
> | walker2d-medium-replay    |        96.0 |  82.8 |  92.4 |  94.42 |
> | halfcheetah-medium-expert |       106.1 |  95.6 |  98.3 |  98.25 |
> | hopper-medium-expert      |       112.5 | 113.4 | 112.6 | 112.11 |
> | walker2d-medium-expert    |       112.3 | 112.1 | 114.5 | 114.62 |
> | antmaze-umaze             |        98.2 |     - |  96.4 |  99.2  |
> | antmaze-umaze-diverse     |        76.0 |  72.6 |  83.4 |  84.8  |
> | antmaze-medium-play       |        86.5 |     - |  82.8 |  85.6  |
> | antmaze-medium-diverse    |        88.7 |  80.2 |  80.8 |  85.6  |
> | antmaze-large-play        |        69.8 |     - |  63.8 |  71.2  |
> | antmaze-large-diverse     |        64.6 |  52.1 |  53.2 |  52.8  |
> | Average (15 tasks)        |        87.0 |    -- |  85.2 |  87.7  |
>
>
> (Q3: On benchmarks based on LIBERO / ManiSkill3)
>
> We thank the reviewer for suggesting newer simulators such as LIBERO and ManiSkill3. These platforms are promising for embodied AI, but to our knowledge they currently do not provide standardized offline RL benchmarks with fixed logged datasets and widely adopted evaluation protocols comparable to D4RL. Recent offline RL methods highlighted by the reviewer (including A2PO and VABC) also conduct their main empirical study on D4RL rather than on LIBERO/ManiSkill-style setups.
>
> For fairness and comparability within the offline RL literature, we therefore follow this common practice and evaluate on the full D4RL suite (Gym, AntMaze, Adroit, Kitchen). Compared to A2PO, we omit Maze2D (point-mass control) and focus on the more challenging navigation and manipulation tasks (AntMaze, Kitchen), where OOD action selection is especially important and ADAC shows clearer gains. We also use a PointMaze environment as a qualitative illustration in the main text. We will clarify this benchmark choice and briefly discuss extending ADAC to future large-scale offline datasets built on LIBERO or ManiSkill3 once suitable offline benchmarks become available.
>
>
> References:
> [1] Williams (1992).
> [2] Sutton et al. (2000).
> [3] Auer et al. (2002).
> [4] “A2PO: Towards Effective Offline Reinforcement Learning from an Advantage-aware Perspective.”
> [5] “Value-aligned Behavior Cloning for Offline Reinforcement Learning via Bi-level Optimization.”
> [6] “LIBERO: Benchmarking Knowledge Transfer for Lifelong Robot Learning.”
> [7] “Demonstrating GPU Parallelized Robot Simulation and Rendering for Generalizable Embodied AI with ManiSkill3.”

---

### Official Review · Reviewer_SCwX · 2025-11-04

**Soundness:** 3
**Presentation:** 3
**Contribution:** 2
**Rating:** 4
**Confidence:** 4

**Summary:**

The paper proposes ADAC (Advantage-based Diffusion Actor-Critic) for offline RL. The main idea is to evaluate potentially OOD actions by how much next-state value they can reach, using a value function learned by expectile regression as a proxy for the dataset-optimal value. Based on this, the authors define an advantage that compares E_{s’}[V(s’)] under candidate action a to a κ-quantile of values from actions sampled from the behavior policy, and then modulate the critic target with A(s,a) in the Bellman backup. Experiments on D4RL show strong averages across Gym, AntMaze, Adroit, and Kitchen, plus a PointMaze study visualizing that ADAC recovers near-optimal, straight-line paths absent from the dataset. Ablations and a brief efficiency comparison are also provided.

**Strengths:**

- Paper is well-written and easy to read.
- A novel formulation of an OOD filter. Defining advantage via next-state value relative to a κ-quantile of behavior actions is simple, tunable, and aligns with the claim that V is often more reliable than Q in offline data. The analysis that expectile regression moves V toward a dataset-optimal value is helpful context.
- Compelling qualitative evidence. The PointMaze visualization clearly shows ADAC stitching suboptimal trajectories and discovering straight-line solutions missing from the dataset.
- Good empirical results. ADAC is competitive or better on most D4RL suites, with large gains on AntMaze, which is where OOD action selection matters most.
- Useful ablations/diagnostics. κ-sensitivity, advantage statistics across tasks, and practical soft-clip for stabilizing extreme A add transparency.

**Weaknesses:**

- Missing prior work, prior work already links OOD action selection to “optimal next-state value.” - POR (Policy-Guided Offline RL) [1], which trains a guide policy toward optimal next states and uses that signal to permit OOD generalization. Please cite and compare against POR.
- Accuracy of A(s,a) hinges on policy and model constraints. Although A(s,a) is intended to promote good OOD actions, its reliability depends on (i) the transition model producing realistic s’ for actions sampled from \pi since extreme OOD actions can yield poorly extrapolated s’; and (ii) the value function’s generalization on OOD states. Errors in either component can make A(s,a) over-optimistic or over-conservative, weakening the claimed advantage-selection benefit.
- System complexity and tuning burden. The method adds several moving parts: an expectile-trained V, a learned dynamics model, and extra hyperparameters (quantile threshold \kappa, expectile \eta, penalty/weight \alpha, sampling counts). This increases implementation complexity and tuning overhead, which may limit practicality and comparability across domains.
- Reporting/variance. Several results show large standard deviations (e.g., AntMaze large variants in Table 1; Adroit pen tasks), and all scores use only 4 seeds. Strong claims (e.g., “SOTA across almost all tasks”) should be tempered or backed with more seeds.

[1] Xu et al, A Policy-Guided Imitation Approach for Offline Reinforcement Learning, NeurIPS 2022.

**Questions:**

1. Why is the A(s,a)-augmented critic provably better than standard TD? Can you provide a formal error decomposition (e.g., showing bias/variance reduction, a tighter fixed-point error bound) that demonstrates Q learned with A(s,a) surpasses vanilla TD under reasonable assumptions?
2. DQL baseline. If my understanding is correct, removing A(s,a) reduces the method to Diffusion Q-learning with your current infrastructure. Please report the results of DQL baseline under the same codebase, architectures, sampling counts, and training/evaluation settings. This would isolate the key benefit of the proposed advantage term.

---

> ### Author Response · Authors · 2025-11-21
>
> (Q1: On theoretical guarantees for the advantage-modulated critic)
>
> Thank you for the question. We would like to clarify that providing a general theorem showing that our augmented critic *always* outperforms vanilla TD is fundamentally difficult under realistic offline RL assumptions. Because the dataset has only partial coverage, the values of truly OOD actions and their Bellman targets cannot be reliably identified, which makes a pointwise comparison between the two operators hard to formalize. Existing analyses of offline RL similarly indicate that, without strong coverage assumptions, value behavior outside the data support is intrinsically hard to characterize.
>
> Moreover, the standard TD operator and our augmented operator have different extrapolation behavior outside the dataset support, so their fixed points are not directly comparable in a universal sense. For these reasons, a broad “superiority” guarantee is not straightforward to state. Instead, we focus on properties that remain meaningful in the partial-coverage regime: our augmented operator is still $\gamma$-contractive and admits a uniform bound on its fixed-point deviation. These are, to the best of our knowledge, the strongest guarantees one can provide without imposing unrealistic coverage assumptions.
>
> (Q2: On the DQL baseline without advantage modulation)
>
> We appreciate the request for a Diffusion Q-learning (DQL) baseline implemented under exactly the same infrastructure. In fact, the ablation in Sec. 5.3 already corresponds to this setting: the “w/o adv” variant is obtained by *only* removing the advantage modulation term $A(s,a)$ from the critic update, while keeping everything else identical (same diffusion policy architecture, value and dynamics networks, training schedule, sampling counts, and hyperparameters). This is precisely the DQL-style baseline the reviewer is asking for.
>
> As reported in Fig. 5, adding the advantage term on top of this DQL baseline consistently improves performance across all four representative domains, with relative gains of about 11% on Gym Locomotion, 12% on AntMaze, 11% on Adroit, and 12% on Kitchen. In the revision, we will make this correspondence explicit by (i) clearly stating that “w/o adv” is our DQL baseline under the same codebase, and (ii) adding a short paragraph in the experimental section summarizing these domain-level improvements to more directly highlight the benefit of the proposed advantage modulation.

---

### Comment · Area_Chair_zfu8 · 2025-11-23
**Action Needed: Follow-Up Assessments Pending**

Dear Reviewers,

The authors have submitted their rebuttal, and we now require your follow-up assessments to move the decision process forward. Please review the authors’ responses and update your evaluations accordingly.

Your prompt follow-up is necessary for us to finalize the meta-review.
Kindly submit your updates as soon as possible.

Best,
Area Chair

---

### Note · Authors · 2026-01-15

I have read and agree with the venue's withdrawal policy on behalf of myself and my co-authors.